

# Synergetic effects of NH₃ and NOx on the production and optical absorption of secondary organic aerosol formation from toluene photooxidation

Shijie Liu [a], Dandan Huang [b], Yiqian Wang [a], Si Zhang [a], Can Wu [a], Wei Du [a], Gehui Wang [a,c,*]

[a] Key Lab of Geographic Information Science of the Ministry of Education, School of Geographic Sciences, East China Normal University, Shanghai 210062, China

[b] State Environmental Protection Key Laboratory of Formation and Prevention of the Urban Air Pollution Complex, Shanghai Academy of Environmental Sciences, Shanghai 200233, China

[c] Institute of Eco-Chongming, 3663 North Zhongshan Road, Shanghai 200062, China

Corresponding author: Prof. Gehui Wang, e-mail: ghwang@geo.ecnu.edu.cn



## Abstract

$NH_3$ is the most important alkaline gas in the atmosphere and one of the key species affecting the behaviors of atmospheric aerosols. However, the impact of $NH_3$ on secondary organic aerosol (SOA) formation remains poorly understood, especially the dynamic evolution of chemical compositions in the SOA formation process. A series of chamber experiments was performed to probe the individual and common effects of $NH_3$ and NOx on toluene SOA formation through OH-photooxidation. The chemical compositions of toluene SOA were characterized using the Aerodyne high-resolution time-of-flight aerosol mass spectrometer (AMS). From $637 \pm 14.6$ µg m$^{-3}$ (control), the SOA mass concentration increased to $867 \pm 12.7$ µg m$^{-3}$ in the presence of $NH_3$ and decreased to $452 \pm 18.9$ µg m$^{-3}$ in the presence of NOx. However, the highest SOA concentration ($1020 \pm 10.6$ µg m$^{-3}$) and the lowest carbon oxidation state ($OS_C$) occurred in the presence of both $NH_3$ and NOx, indicating that the higher volatility products that formed in the presence of NOx could precipitate into the particle-phase when $NH_3$ was added. This resulted in a synergetic effect on SOA formation when $NH_3$ and NOx co-existed. The heterogeneous reaction was the main pathway by which $NH_3$ participated in SOA formation in the photooxidation process. The synergetic effect of $NH_3$ and NOx was also observed in SOA optical absorption. A peak at 280 nm, which is characteristic of organonitrogen imidazole compounds, was observed in the presence of $NH_3$ and its intensity increased when NOx was added into the chamber. This work improves our understanding of how the synergistic interactions between $NH_3$ and NOx influence SOA formation and offers new insights into mitigating aerosol pollution that





factor in mixed atmospheric conditions.

**Keywords**: Photooxidation; Toluene; $NH_3$; Dynamic characteristics; Synergistic
effects


# 1 Introduction


Secondary organic aerosols (SOA) are an important component of atmospheric
particulate matter (Moise et al., 2015; Liu et al., 2017). SOA can significantly affect
atmospheric visibility, air quality, and subsequently, public health (Paciga et al., 2014;
Yang et al., 2016; Liu et al., 2017). The optical properties of aerosols have been directly
and indirectly linked to their effects on the climate (Laskin et al., 2015; Xie et al., 2017;
Peng et al., 2020). Because of the complexity of their chemical components, oxidation
processes, and environmental factors, SOA formation mechanisms are very complex
and the current understanding of SOA formation is incomplete. This limited
understanding hampers the ability of models to predict the magnitudes, dynamics, and
distributions of atmospheric aerosols from particulate and precursor emissions (Ortiz-
Montalvo et al., 2014). In the past decades, although our understanding of SOA
formation mechanisms has been constantly improving, there is still a gap between the
simulated SOA concentration in large-scale atmospheric models and field observations
(Volkamer et al., 2006; Yang et al., 2018).
Ammonia ($NH_3$) is the most important alkaline inorganic gas, it is widespread in
the atmosphere and is one of the critical factors influencing SOA formation (Wang et
al., 2018; Chen et al., 2019). Some studies have noted that the presence of $NH_3$ can
contribute to the formation of more aerosol mass through photooxidation (Na et al.,
2007; Li et al., 2018). Na et al. (2007) observed that aerosol yields in the α-pinene-
ozone oxidation system increased by 8% when $NH_3$ was added. Li et al. (2018)
concluded that the presence of $NH_3$ in the aromatic hydrocarbon photooxidation system



increased aerosol size growth potentials (by 7%–108%), and resulted in enhanced SOA
formation. Qi et al. (2020) found that the concentration and average diameter of SOA
showed an immediate and rapid increase after adding $NH_3$. Furthermore, the acid-base
reactions between $NH_3/NH_4^+$ and the carboxyl groups in SOA molecules might enhance
SOA formation (Qi et al., 2020; Liu et al., 2015). The condensable ammonium salts
formed from the reaction between $NH_3$ and organic acids reduce the volatility of the
organic acids by several orders of magnitude (Paciga et al., 2014), and act as particle-
phase organics that further promote SOA formation (Na et al., 2007; Huang et al., 2012;
Chen et al., 2019; Qi et al., 2020). Along another pathway, glyoxal can undergo
nucleophilic attack by $NH_3$ through the Maillard reactions and form the corresponding
iminium intermediates (Nozière et al., 2009; Laskin et al., 2015; Liu et al., 2015). The
iminium intermediates can continue to react with carbonyls, which activates further
transformations such as the formation of heterocyclic compounds and oligomerization
reactions and forms condensation (oligomeric) products with more stable secondary
imines (Schiff bases) (Laskin et al., 2014). Both Nozière et al. (2009) and Ortiz-
Montalvo et al. (2014) reported $NH_3$ is an efficient catalyst for reactions with carbonyl
compounds to form nitrogen-containing compounds (NOCs). The reaction between
carbonyl and $NH_3$ can significantly decrease the volatility of oxidation products, which
further increases the yield of SOA (Lee et al., 2013; Zhang et al., 2015; Qi et al., 2020).
Babar et al. (2017) found that the substantial formation of secondary imines in the
presence of $NH_3$ was responsible for the higher α-pinene SOA yields. However, not all
studies have shown that the presence of $NH_3$ increases SOA yields. One study observed



that $NH_3$ suppressed SOA formation under certain ozonation conditions (Ma et al.,
2018b). Furthermore, the consumption of $NH_3$ by Criegee intermediates was reported
to decrease the secondary ozonide yield and thus affect SOA formation.
Nitrogen oxides (NOx = NO + $NO_2$), which are mainly emitted from the
combustion of fossil-fuels, have received significant attention due to their effects on the
photooxidation process of volatile organic compounds (VOCs) and SOA formation
(Draper et al., 2015; Berkemeier et al., 2016; Zhao et al., 2018; Surratt et al., 2006; Ng
et al., 2007b; Sarrafzadeh et al., 2016). Laboratory experiments have found that SOA
formation was initially enhanced, but then suppressed with increasing NOx
concentrations (Sarrafzadeh et al., 2016; Yang et al., 2020). The competitive chemistry
of organic peroxy radicals ($RO_2$) with hydroperoxyl radicals ($HO_2$) and NO was
responsible for the variability in SOA formation (Xu et al., 2014; Ng et al., 2007a; Jiang
et al., 2020). $RO_2$ mainly reacts with $HO_2$ under low-NOx conditions to form oxidation
products with lower volatility, which may enable it to participated into the particle-
phase and contribute to the SOA mass (Ng et al., 2007a). While the $RO_2$ + NO reaction
is predominant in high-NOx conditions, the increase in volatile products formed
through fragmentation was responsible for the decrease in SOA yield with increasing
NOx (Zhao et al., 2018; Liu et al., 2019a; Xu et al., 2020). In addition, the suppressing
effect of NOx on OH concentrations was another reason for the decreasing trend in
SOA yields under high-NOx conditions (Sarrafzadeh et al., 2016).
In recent decades, atmospheric pollutants in China have changed significantly in
their concentrations and composition (Wang et al., 2015; Xia et al., 2016). China



decreased the emissions of $SO_2$ and NOx by 75% and 10% from 2007–2015 and from
2011–2015, respectively (de Foy et al., 2016; Vu et al., 2019; Wang et al., 2020). Owing
to the lack of regulation regarding $NH_3$ emissions, $NH_3$ emissions increased by ~30 %
from 2008–2016 over the North China Plain (Liu et al., 2018). As has been pointed out
in previous research, the effect of $NH_3$ on the formation of SOA is one of gradual
enhancement and may counteract any decreases in SOA formation due to reductions in
$SO_2$ and NOx (Zhang et al., 2021). Indeed, one study observed that a reduction of $NH_3$
emissions improved $PM_{2.5}$ pollution compared to $SO_2$ in winter (Erisman and Schaap,
2004). Hence, the mechanism by which $NH_3$ affects SOA formation has attracted more
and more attention. However, previous studies have not paid sufficient attention to the
joint impacts of $NH_3$ and NOx on the formation of SOA and its corresponding optical
properties. Due to the lack of real time detection methods for SOA chemical
composition, the dynamic characteristics of how $NH_3$ participates in SOA formation
via photooxidation have not been extensively studied.
Toluene is one of the most abundant aromatic VOCs in the urban atmosphere,
which is also an important source of BrC (Ma et al., 2018a; Laskin et al., 2010). The
effects of $NH_3$ and NOx on SOA formation through the toluene photooxidation process
were investigated in this study. The chemical composition of toluene SOA was
characterized on-line with an aerosol mass spectrometer and the dynamic
characteristics of SOA chemical composition under different conditions were further
explored by applying a positive matrix factorization (PMF) analysis. The optical
properties of toluene SOA particles were determined based on a UV-vis spectrum



analysis. Possible mechanisms of the effects of both $NH_3$ and NOx on SOA formation
were discussed. The results will help us to better understand SOA formation
mechanisms in complex pollution conditions with elevated $NH_3$ and NOx
concentrations in an urban atmospheric environment.

## 2 Materials and Methods


### 2.1 Photooxidation chamber experiments


All toluene photooxidation experiments were performed in a 4 $m^3$ chamber. The
chamber has been described in detail in our previous studies. Briefly, the chamber was
constructed with a 0.08 mm-thick FEP-Teflon film. The average particle wall-loss rate
constant of $3.6 \times 10^{-5}$ $s^{-1}$ was used to correct the measured particle concentrations and
SOA yields in this study. 50 UV-B lamps (TUV36W, Philips) with peak wavelengths
of 254 nm were set up around the chamber and used as the light source to drive OH
radical formation through hydrogen peroxide ($H_2O_2$) photolysis. Mirror surfaced
stainless steel was used as the interior wall of the enclosure to maximize and
homogenize the interior light intensity.
Before each experiment, the chamber was flushed with zero air for at last 18 hours,
after which the concentration of particles was less than 1 $cm^{-3}$. Zero air was generated
by a zero air supply (111-D3N, Thermo Scientific[TM], USA). The flow rate of zero air
was controlled at 20 L $min^{-1}$ by a mass flow controller (D088C/ZM, Beijing Sevenstar
Electron Corporation) during the process of inflating. The relative humidity (RH) of



zero air was about 15~20%. For each experiment, measured amounts of toluene
(Sigma-Aldrich, analytically pure) and $H_2O_2$ solution (Sigma-Aldrich, 30 wt% in $H_2O$)
were injected into a Teflon bulb with micro syringes. Zero air was passed through the
injection tube to make sure all the liquids had evaporated to the gas-phase and were
blown into the chamber. NOx (Air Liquid Shanghai, 510 ppm $NO_2$ in $N_2$) and $NH_3$ (Air
Liquid Shanghai, 502 ppm $NH_3$ in $N_2$) were introduced directly into the chamber to
reach the required concentrations. For experiments with NOx, although only $NO_2$ was
introduced into the chamber before photooxidation, NO could be formed through $NO_2$
photolysis under the UV light irradiation, so NO always coexisted with $NO_2$ in the
photooxidation system (Zhao et al., 2018). Each experiment was performed without
seed aerosols present. The experimental conditions for the toluene photooxidation are
listed in Table 1.
**2.2 Particle concentration measurements**

For each experiment, a scanning mobility particle sizer (SMPS) was used to record

the particle size distribution and volume concentration of the toluene-derived SOA. The
SMPS was composed of a differential mobility analyzer (DMA model 3081, TSI Inc.,
USA) and a condensation particle counter (CPC model 3776, TSI Inc., USA) which
were used for screening particles with specific aerodynamic equivalent sizes and for
counting the number of the selected particles, respectively. The sheath gas velocity was
3 L min$^{-1}$ and the sample gas velocity was 0.3 L min$^{-1}$. The scan was repeated every 5
min. During each scan circle, the scan time was 240 s, and the particle sizes ranged



from 13.6 nm to 726.5 nm. A density of 1.4 g m$^{-3}$, which was measured by Ng et al.
(2007), was used for the calculation of toluene SOA mass concentration from the
particle volume concentration (Ng et al., 2007b).

## 2.3 Chemical characterization

In this study, the toluene SOA chemical compositions were characterized with an
on-line high-resolution time-of-flight aerosol mass spectrometer (HR-ToF-AMS,
Aerodyne Research Inc. USA). The sample flow passed through a Nafion dryer and the
RH of the sample gas was reduced to below 20% before entering the AMS. In the
injection port, an aerodynamic lens focused particles with a vacuum aerodynamic
diameter below 1 μm into a narrow beam. Particles impacted a flash vaporizer (600°C)
at the rear of the sizing region under high vacuum (~10$^{-7}$ Torr) and were subsequently
ionized by electron impact ionization (70 eV). Then, the positively charged ions entered
the ToF section and were separated. V-mode (m/Δm = ~2000) was used in the AMS
ToF section to address the high signal-to-noise. The separated ion fragments were
analyzed by a quadrupole mass spectrometer with scans from 1 to 300 m/z. The
composition-dependent collection efficiency (CE) was applied to the data based on the
methods established by Middlebrook et al. (2012). For mass concentration calculations,
1.1, 1.2, and 1.4 were applied as the default relative ionization efficiency (RIE) values
of nitrate, sulfate, and organic compounds, respectively. The standard AMS data
analysis software SQUIRREL 1.63B coupled with PIKA 1.23B in the Igor Pro
(WaveMetrics,    Inc.,    Portland,    Oregon),    which    were    retrieved    from



http://cires1.colorado.edu/jimenez-group/ToFAMSResources/ToFSoftware/, were used
for the analysis of elemental ratios and the ion speciated compositions of toluene SOA
in the chamber. Note that the elemental ratios (i.e., O/C, H/C, and N/C) and mass-to-
carbon ratio (OM/OC) were all calculated using the Aiken-Ambient method for
comparability with previous studies (Aiken et al., 2008). In order to further explore the
changes in SOA chemical composition, a PMF of the high-resolution mass spectra was
performed to determine the different organic aerosol (OA) factors during the toluene
photooxidation process. We preformed the PMF analysis in the same way as Zhang et
al. (2011), the details of which are provided in the supporting information.
**2.4 Absorption measurements**
The changes of absorption spectra and the absorbance of the toluene derived SOA
under different conditions were determined using a UV spectrophotometer (UV-3600,
Shimadzu, Japan) with a 1 cm cuvette. The SOA was collected from a 3 $m^3$ sample gas
onto the 46.2 nm PTFE filter (Whatman$^{TM}$, UK). The collected SOA sample was
dissolved in 5 mL of methanol (HPLC grade, > 99.8%) with 30 min of sonication. The
filter extracts were filtered through 0.2 μm PTFE syringe filters to remove suspended
insoluble particles. Before detection of the optical absorbance, a cuvette filled with pure
methanol was scanned as a blank to provide a spectral background. The absorption was
detected over the range of 200 to 800 nm with a resolution of 0.5 $nm^{-1}$. The light
absorption coefficient of the particles at a specific wavelength λ ($Abs_\lambda$, M/m) was
calculated according to Eq. R1:





$$Abs_{\lambda}=(A_{\lambda}-A_{700}) \cdot \frac{V_l}{V_a \cdot L} \cdot \ln(10) \qquad (R1)$$

where, $A_{700}$ is the background value of light absorption intensity, calculated as the
average value of light absorption intensity from 695–705 nm to reduce the limits of
error in measurement; $V_l$ and $V_a$ are the volumes of methanol with dissolved particles
and sampled air, respectively; and L is the optical path length. Because $Abs_{\lambda}$ was
strongly dependent on the amount of SOA, all $Abs_{\lambda}$ results were normalized based on
the SOA mass collected on the filter. The normalized result was defined as the mass
absorption coefficient (MAC, $m^2$ $g^{-1}$), calculated using Eq. R2:

$$MAC_{\lambda}=\frac{Abs_{\lambda}}{M} \qquad (R2)$$

where, M ($\mu g$ $m^{-3}$) represents the concentration of methanol-soluble organic carbon.

## 3 Result and Discussion

### 3.1 SOA formation

In order to investigate the effect of $NH_3$ and $NOx$ during SOA formation from
toluene photooxidation, a control test was carried out. The SOA mass concentrations at
different conditions are shown in Fig. 1. There was a noticeable increase in the SOA
mass concentration in the presence of $NH_3$. The mass concentration of SOA increased
from $637 \pm 14.6$ $\mu g$ $m^{-3}$ without $NH_3$ to a maximum of $867 \pm 12.7$ $\mu g$ $m^{-3}$ with 200 ppb
$NH_3$, consistent with previous studies (Na et al., 2007; Qi et al., 2020). Previous studies
attributed the enhancement of SOA to the formation of NOCs from acid-base reactions
between $NH_3/NH_4^+$ and carboxyl groups, or Maillard reactions of $NH_3/NH_4^+$ with





carbonyl functional groups (Nozière et al., 2009; Ortiz-Montalvo et al., 2014; Liu et al.,
2015; Qi et al., 2020). In contrast, SOA concentrations were lower in the presence of
NOx, and the maximum mass concentration of toluene SOA was only $452 \pm 18.9$ μg m$^{-}$
$^3$ with 63 ppb initial NOx. Numerous studies have shown that, instead of $RO_2$ reacting
with $RO_2/HO_2$, NO would react with $RO_2$ to form the RO intermediate and produces
more oxidation products with higher volatilities through fragmentation in the presence
of NOx (Zhao et al., 2018; Liu et al., 2019a; Xu et al., 2020). Highly volatile compounds
cannot readily participate into the particle-phase, so this substantially suppresses the
formation of SOA.

The NOx and $NH_3$ had opposite effects on toluene SOA formation in this study.

Interestingly, however, the highest SOA mass concentration ($1020 \pm 10.6$ μg m$^{-3}$)
occurred in the presence of both NOx and $NH_3$, which was nearly 1.6 times higher than
that observed with no NOx or $NH_3$. Therefore, together $NH_3$ and NOx had a synergistic
effect on SOA formation because their combined effect on SOA formation was greater
than the sum of their separate effects. This may explain why predictions of SOA
concentrations in large-scale atmospheric models, which typically describe SOA
formation from data derived from chamber experiments, are frequently lower than field
observations (Volkamer et al., 2006). The effects of multiple factors are not well-
characterized by chamber experiments, which was partly responsible for the gap
between the simulations and field observations.



## 3.2 SOA chemical composition


The traditional SOA formation mechanism is based on the chemical compositions
obtained through off-line detection of the chemical composition of SOA (Jang et al.,
2002; Liu et al., 2019a; Xu et al., 2020). SOA is dynamic and continually evolving in
the atmosphere and the ageing process of SOA co-occurs with its formation process.
Hence, the transformation of the SOA chemical composition during the SOA formation
process is generally believed to be widespread, but is rarely characterized in the
previous studies. Therefore, the AMS was used for on-line measurement of the SOA
chemical composition and how the chemical composition evolved in the photooxidation
process would be discussed in this section below.
The chemical composition of SOA is very complex. The average carbon oxidation
state ($OS_C$) has been shown to be an ideal conceptual framework to describe changes
in the degree of oxidation undergone by SOA (Kroll et al., 2011), and has been widely
applied in field and laboratory studies (Chen et al., 2018; Mandariya et al., 2019). $OS_C$
is calculated based on the measurements of O/C and H/C ($OS_C \approx 2 \times$ O:C - H:C). Fig.
2 shows the changes in the $OS_C$ of toluene SOA formed in different experiments.
Notably, all the toluene SOA was characterized as semi-volatile oxygenated organic
aerosols (SV-OOA) with $OS_C$ values ranging between -0.5 and 0. However, the
different $OS_C$ values and the change trends observed for the toluene SOA formed in
different conditions (with and without $NH_3$/NOx) in Fig. 2 indicated that there was a


photooxidation mechanism active during SOA formation, which ultimately changed the
SOA chemical compositions.
The $OS_C$ increased over time for all SOAs that were formed in the absence of $NH_3$.
There are several possible reasons for the increasing trend of $OS_C$ values. Firstly, a
dynamic equilibrium of semi-volatile vapors may have been achieved between the
particle-phase and gas-phase during the earlier toluene oxidation process. The increase
of SOA led to a reduction in the concentration of gas-phase semi-volatile organic
products. A decreasing concentration of gas-phase semi-volatile organic compound
products would suppress their transformation from gas-phase to particulate-phase.
More lower volatility gas-phase oxidation products with higher $OS_C$ values would then
be shifted toward to the particle phase, which would be responsible for the continuing
increase of SOA. Secondly, the formed SOA could have been further oxidized by OH
through heterogeneous reactions (Kourtchev et al., 2015; Liu et al., 2019b). This could
be the main reason for the increase in the $OS_C$ when the SOA concentration was no
longer increasing. Finally, even there was no OH in the chamber, the photodegradation
of SOA can produce oxygenated volatile organic compounds (OVOCs, e.g. acetic acid,
acetaldehyde, and acetone) under UV light irradiation, potentially leading to
measurable mass loss from SOA (Malecha and Nizkorodov, 2016). The
photoproduction of OVOCs from SOA had a lower $OS_C$ than that of SOA, therefore,
the $OS_C$ of SOA had increased to a certain extent.
The fact that additional photochemical processing results in the dynamic evolution
of the $OS_C$ over time has been demonstrated in both field and laboratory experiments





(Jimenez et al., 2009). The atmospheric oxidation of OA tends towards higher $OS_C$
regardless of the original OA source (Herndon et al., 2008). However, when $NH_3$ was
present, the $OS_C$ of total SOA went almost unchanged for the whole photooxidation
period. Carboxy and carbonyl are the main oxygen-containing functional groups
responsible for the toluene photooxidation production (Ji et al., 2017). An organic
ammonium salt with four H atoms can offset an increase caused by the formation of
organic acids/carboxy group through acid-base reactions (Kuwata and Martin, 2012;
Liu et al., 2015). Or $NH_3/NH_4^+$ may react with carbonyl functional groups through
Maillard reactions, consuming the oxygen in the carbonyl group and leading to the
formation of species with covalent carbon-nitrogen bonds (Lee et al., 2013; Zhang et
al., 2015; Qi et al., 2020). Xu et al. (2018) showed that imidazole compounds (OSC ≈
-1.3) generated through heterogeneous reaction between $NH_3$ and carbonyl compounds
might contribute to the decrease in the $OS_C$ of SOA. It is clear that an increase in $OS_C$
caused by the formation of oxygen-containing functional groups (e.g., carboxy,
carbonyl, etc.) would be counteracted through acid-base reactions or Maillard reactions
in the presence of $NH_3$. After 60 min of UV light irradiation, there was no more SOA
formation; however, the $OS_C$ did decrease slightly, illustrating that the $NH_3$ could
continue to react with SOA through heterogeneous processes. Huang et al. (2016) also
pointed out that the portion of semi-volatile products with low $OS_C$ formed at the later
stage of photooxidation also contributed to the decreased $OS_C$.

The $OS_C$ of the toluene SOA formed with NOx was lower than that formed in the

absence of NOx, no matter whether $NH_3$ was present in the chamber or not. This





indicated that an increased NOx concentration benefits the formation high volatility
oxidation products with lower $OS_C$ values (Kroll et al., 2011; Jimenez et al., 2009).
However, the relationships between $OS_C$ and SOA mass concentration with and without
$NH_3$ were the opposite of each other. Predictably, the SOA formation mechanism in the
presence of NOx is different from that with NOx + $NH_3$. In the absence of $NH_3$, the RO
intermediate, which is easily fragmented to produce relatively high-volatility
compounds, was the dominant product of the NOx + $RO_2$ reaction (Zhao et al., 2018;
Liu et al., 2019a; Xu et al., 2020). Highly volatile compounds cannot readily precipitate
into the particle-phase, which subsequently results in a lower SOA yield in the presence
of NOx (Yang et al., 2020). Thereby, both $OS_C$ and the SOA mass concentration were
lower when 60 ppb NOx was added into the chamber. However, when both NOx and
$NH_3$ were present, the toluene derived SOA had the lowest $OS_C$ value, but the highest
mass concentration. This result suggested that although NOx promotes the formation
of higher volatile compounds, these higher volatile compounds (e.g. glyoxal) can react
with $NH_3$ and precipitate into the particle-phase, which could contribute to the increase
in SOA formation. Huffman et al. (2009) observed that aerosol volatility was inversely
correlated with the extent of oxidation of OA components. The low value of $OS_C$ in the
presence of NOx indicated that NOx would promote the formation of the relatively
high-volatility compounds. However, the lower $OS_C$ value in the presence of $NH_3$
indicated that the high-volatility compounds would promote precipitation into the
particle-phase when reacting with $NH_3$.

Fragments derived from the AMS data have also been widely used to explore the


bulk compositions and properties of SOA (Ng et al., 2010; Ng et al., 2017). The m/z 43
(f43) frequency was dominated by ion $C_2H_3O^+$, which is the tracer for organic
compounds with alcohol and carbonyl functional groups (Alfarra et al., 2006).
Meanwhile, the m/z 44 (f44) signal was dominated by $CO_2^+$ ions, which is the tracer
for organic compounds with carboxyl functional groups and indicator of highly
oxygenated organic aerosols (Ng et al., 2010). Here, we used f43 and f44 to express the
fractions of $C_2H_3O^+$ and $CO_2^+$ to the total organic signal. The change of f43 *vs.* f44,
which has an inflection point during the photooxidation process, is shown in Fig. 3. In
our study, the change before the inflection point was defined as the formation stage,
and the linear fit of f43 *vs.* f44 for the formation stage is shown by the dashed lines.
The change in f43 *vs.* f44 after the inflection point was defined as the stable stage, and
the linear fit of f43 *vs.* f44 in this stage is shown by the solid lines. The formation and
stable stages of the f43 *vs.* f44 relationship during the experiment are discussed
separately here.
In the stable stage, the increase in f44 and decrease in f43 with increasing OH
exposure indicated that the carbonyl groups in toluene SOA were oxidized to carboxyl
groups by the ageing process. For the experiments without $NH_3$ and NOx, the slope
ratio of f43 *vs.* f44 was -3.9. When there was 60 ppb initial NOx, the f43 was almost
stable while the f44 increased with the oxidation process. There was a lower slope ratio
of f43 *vs.* f44, indicating that organic compounds with more alcohol and carbonyl
functional groups had formed in the presence of NOx. But for the experiments with 200
ppb initial $NH_3$, the slope ratios of f43 *vs.* f44 were only -1.1 and -1.3 in the presence





and absence of NOx, respectively. According to the above results, we can see that more
carbonyl groups are consumed as carboxyl groups are formed in the presence of $NH_3$.
The carbonyls can be oxidized to organic acids (Kawamura and Bikkina, 2016), but
extra-consumed carbonyls can be nucleophilically attacked by $NH_3/NH_4^+$ to form
nitrogen-heterocyclic compounds, e.g., imidazole (Grace et al., 2019; Lian et al., 2020).
Meanwhile, the peak f44 value decreased from 0.13 to 0.10 when $NH_3$ was added into
the chamber. This suggested that the heterogeneous reaction of $NH_3/NH_4^+$ could
promote the consumption of particle-phase carbonyl groups (Xu et al., 2018), and must
inhibit the formation of carboxyl groups in the SOA ageing process. The differences in
spectra of toluene SOA in the formation stage and stable stage are shown in Fig. 4. A
lower signal intensity variation of $CO_2^+$ in the presence of $NH_3$ also illustrated that $NH_3$
would inhibit heterogeneous reactions that form carboxyl groups.
In the formation stage, the slope ratios of f43 *vs.* f44 were almost the same for both
experiments without NOx. It can thus be seen that the presence or absence of $NH_3$ does
not affect the change trend of f43 *vs.* f44 in the SOA formation stage. Therefore, the
gas-phase homogeneous reaction of $NH_3$ on SOA formation is not important. Clearly,
the particle-phase heterogeneous reaction was the main reaction pathway by which $NH_3$
participated in the photooxidation process and toluene SOA formation. However,
negative correlations were observed between f43 and f44 in the presence of NOx. Based
on this, we concluded that NOx not only affects the SOA formation through the particle-
phase heterogeneous reactions, but also through gas-phase homogeneous reactions.


## 3.2 PMF results


A temporal evaluation of the toluene SOA chemical composition during
photooxidation is vital to the analysis of the NOA formation mechanism in the presence
of $NH_3$ and/or NOx. Therefore, this study further compared the chemical properties of
the SOA generated under different experimental conditions by applying a PMF analysis
to the HR-ToF-AMS data (Chen et al., 2019). A summary of the PMF results is
presented in Fig. S1-S4. For the toluene OH-photooxidation experiments where NOx
or $NH_3$ were present, the PMF analysis identified two factors. High-nitrogen OA (Hi-
NOA) was tentatively assigned to the high N/C values and low-nitrogen OA (Lo-NOA)
to the low N/C values. Fig. 5 compares the H/C, O/C, and N/C values of Hi-NOA and
Lo-NOA in each experiment. Fig. 6 exhibits the evolution of Hi-NOA and Lo-NOA
during the photooxidation process as resolved from the PMF analysis of different initial
NOx/$NH_3$ concentrations. While similar evolutionary trends were observed under
different conditions, the relative intensities and the chemical compositions of these two
factors in each experiment were not consistent.
For the toluene SOA formed under $NH_3$ conditions, both Lo-NOA and Hi-NOA
had similar O/C values, which were fully oxygenated with an average of $0.74 \pm 0.04$
(Fig. 5a). These O/C values were comparable to the low-volatility oxygenated organic
aerosols (LV-OOA) with an O/C value ranging from 0.6 to 1 (Jimenez et al., 2009). The
main difference between these two OA sources was the N/C ratio. The N/C ratio of Hi-
NOA (N/C = 0.032) was about three times higher than that of Lo-NOA (N/C = 0.010)



(Fig. 5a). The evolution of these two OA sources during the photooxidation process is
shown in Fig. 6a. The components of toluene SOA were mostly Lo-NOA during the
initial phase of SOA formation, but Hi-NOA toluene SOA started forming after 10
minutes and continued to gradually increase. It was likely that the formation pathway
of Hi-NOA did not involve the reaction of $NH_3$ with organic matter in the homogeneous
gas phase. The Lo-NOA reached the maximum mass concentration after 30 min of
photooxidation, and then decreased. The trends of these two OA factors suggested that
the formed Lo-NOA was converted into something else in the particle-phase. As the
Lo-NOA decreased, the mass concentration of Hi-NOA gradually increased. Thus, the
Hi-NOA should be derived from the heterogeneous reaction of Lo-NOA with
$NH_3/NH_4^+$. With the gradual replacement of Lo-NOA by Hi-NOA, the ratio of [Hi-
NOA]/[Lo-NOA] stabilized at 5~6.

For the toluene SOA formed under NOx conditions, there was not a large

difference between the N/C ratios of Hi-NOA (N/C = 0.019) and Lo-NOA (N/C = 0.014)
(Fig. 5c). At the end of the NOx experiment, the ratio of [Hi-NOA]/[Lo-NOA] was only
3:2 (Fig. 6c). It follows that the contribution of the heterogeneous NOx reaction to the
N/C ratio of toluene SOA was not obvious. Therefore, the formation of NOCs in the
presence of NOx mainly occurred through gas-phase homogeneous reactions, which
was consistent with the results in section 3.3.

The changing trend of N/C with time in the presence of $NH_3$ was different to that

with NOx present. The evolutions of the N/C of SOA in different experiments are
shown in Fig. 7. In the presence of $NH_3$, the N/C value gradually increased throughout





the photooxidation process. The increased N/C value in the photooxidation process was
attributed to the heterogeneous $NH_3$ reaction with SOA. But in the presence of NOx,
the N/C increased rapidly to its maximum value where it was stable for the rest of the
reaction. This could mean that the heterogeneous reaction of toluene SOA with NOx to
form NOCs was not as important as the gas-phase homogeneous reaction.

When both NOx and $NH_3$ were added into the chamber, the N/C ratios of Hi-NOA

and Lo-NOA were 0.062 and 0.29, respectively (Fig. 5b). The N/C ratio of Hi-NOA,
which was comparable to the recently isolated nitrogen-enriched OA value (0.053)
observed by Sun et al. (2011), was much higher than that observed in the experiments
with only $NH_3$ or NOx. It was even higher than the sum of the N/C ratios from both
Exp. 2 with $NH_3$ and Exp. 4 with NOx. In order to calculate the relative contributions
of $NH_3$ and NOx to N/C, it was assumed that the effects of $NH_3$ or NOx on the N/C
ratio in the Hi-NOA and Lo-NOA factors did not change among different experimental
conditions. For Lo-NOA, the contributions of $NH_3$ and NOx to the N/C value were
0.0126 and 0.0164, and their relative intensities were 43% and 57%, respectively. While
for the Hi-NOA, the contributions of $NH_3$ and NOx to the N/C values were 0.0404 and
0.216, and their relative intensities were 65% and 35%, respectively. For the experiment
with both $NH_3$ and NOx, the contribution of $NH_3$ to N/C was higher by 26%, and the
contribution of NOx to N/C was higher by 17% compared to the experiments with
single pollutants. The co-existence of $NH_3$ and NOx further enhanced the N/C value of
toluene SOA, indicating that a synergetic interaction between $NH_3$ and NOx further
enhanced organic nitrogen formation.





### 3.4 Optical absorption


The optical characteristics of toluene SOA formed from different $NH_3$ and NOx
conditions were investigated. The MAC of toluene derived SOA detected over the range
of 200–600 nm are displayed in Fig. 8. Over the entire UV detection range, an increase
in light absorption was observed when the toluene SOA formed in the presence of NOx
or $NH_3$.
Looking at Fig. 8 in detail, we see that the MAC of toluene SOA formed with (red
line) and without (black line) $NH_3$ overlapped at 250 nm, but when the UV wavelength
exceeded 250 nm the MAC of the toluene SOA formed in the presence of $NH_3$ was
higher. The red line reflects an obvious characteristic absorption peak at 270~280 nm,
which was mainly due to the absorption of the $n \rightarrow \pi^*$ electronic transitions. The
imidazole compounds were formed through the Maillard reactions between $NH_3/NH_4^+$
with carbonyl functional groups (Zhang et al., 2015). The C=N double bonds in the
organonitrogen imidazole compounds can act as effective chromophores since both
$\pi \rightarrow \pi^*$ and $n \rightarrow \pi^*$ transitions are chromatically active (Nguyen et al., 2013). The
UV/visible spectrum of imine and pyrrole show broad bands at 270 nm (NIST, 2020),
which was consistent with the UV absorption peak of the $n \rightarrow \pi^*$ band observed here.
According to the AMS results, carbonyl was the main functional group of toluene SOA.
The emergence of absorption peaks at 270~280 nm demonstrated that some
organonitrogen imidazole compounds (e.g. imines and pyrrole) were formed through
the heterogeneous reaction of toluene with $NH_3$. Meanwhile, the high-molecular weight



nitrogen-containing organic species might have formed through Maillard reactions in
the particle-phase (Wang et al., 2010). This was also a reason for the increase in SOA
mass concentration in the presence of $NH_3$.

The green line in Fig. 8 represents the MAC value of toluene-derived SOA in the

presence of NOx, which was also higher than the black line (control) throughout the
UV detection range. When compared with the red line, the green line had no obvious
characteristic peak at 280 nm, but it had higher absorbance in the range between 240
and 280 nm. This indicated that both NOx and $NH_3$ increased the absorbance of toluene
SOA, while the chromophores generated from the reactions between toluene-derived
SOA with either $NH_3$ or NOx did not behave in the same way.

The blue line in Fig. 8 represents the absorbance of toluene SOA formed in the

presence of both NOx and $NH_3$. The MAC of toluene SOA formed in the presence of
both NOx and $NH_3$ was higher than the toluene SOA formed in the presence of either
$NH_3$ or NOx. There might have been a synergetic effect between NOx and $NH_3$ on the
absorbance of toluene SOA. Considering that the mass concentration of toluene SOA
formed in the presence of both $NH_3$ and NOx was the highest, as described in section
3.1, the co-existence of $NH_3$ and NOx may also result in the toluene SOA having
stronger light absorption and atmospheric radiative forcing. We also noted a higher
MAC value at 280 nm, which illustrated that the presence of NOx could promote the
formation of imines and pyrrole in the photooxidation system of toluene with $NH_3$.



## 4 Conclusion

Here we present the results of a study in which we characterized the mass concentrations, chemical compositions, and optical properties of SOA formed from the photooxidation of toluene under different $NH_3$ and $NO_x$ conditions. When compared with the control experiment, the SOA mass concentration data showed that the formation of toluene-derived SOA was enhanced in the presence of $NH_3$, through acid-base reactions between carboxyl groups or Maillard reactions with carbonyl compounds, but inhibited in the presence of $NO_x$. Meanwhile, the mass concentration of toluene SOA formed in the presence of both $NO_x$ and $NH_3$ was higher than those formed under either $NH_3$ or $NO_x$ alone. This result indicated that there was a synergistic interaction between $NH_3$ and $NO_x$ that further enhanced toluene-derived SOA formation. At the same time, the lowest $OS_C$ value was obtained when both $NH_3$ and $NO_x$ were present. We concluded that high volatile compounds, which were formed from toluene photooxidation in the presence of $NO_x$, could react with $NH_3$ to form products with lower volatilities, and promoted the participation of these products into the particle-phase.

Synergetic effects of $NH_3$ and $NO_x$ on the formation of NOCs and the optical properties of SOA were also observed in this study. The heterogeneous reaction was responsible for the formation of NOCs in the presence of $NH_3$; meanwhile, an absorption peak at 270~280 nm, which is characteristic of imine and pyrrole, was observed. In contrast, the formation of NOCs caused by $NO_x$ alone was mainly due to a gas phase homogeneous reaction.



In the actual atmosphere, NOx and NH$_3$ co-exist. Therefore, the findings presented
here clearly show that the synergetic effects of NOx and NH$_3$ should not be neglected.
In the meantime, our work provides a scientific basis for the consideration of synergistic
emission reductions of NH$_3$ and NOx under the compound pollution conditions, which
will contribute to reducing the burden of aerosols in the atmosphere.

## Data availability

The datasets are available upon request to the corresponding authors.

## Author contributions

SL designed the experiment, conducted the experiments, performed the data
interpretation, and wrote the paper. DH performed the data interpretation and wrote the
paper. GW wrote the paper. YW, SZ, CW, WD contributed to the paper with useful
scientific discussions or comments.

## Competing interests

The authors declare that they have no conflict of interest.

## Acknowledgements

This work was financially supported by National Key Research and Development
Plan programs (Grant No. 2017YFC0212703); National Natural Science Foundation of





China (Grant No.41773117, 42005088); the China Postdoctoral Science Foundation
(Grant No. 2019M661427); Fundamental Research Funds for the Central Universities,
Director's Fund of Key Laboratory of Geographic Information Science (Ministry of
Education), East China Normal University (Grant No. KLGIS2021C02); amd ECNU
Happiness Flower Program.

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





## Tables

**Table 1.** Summary of experimental conditions in this study.

| No. | Toluene (ppb) | $H_2O_2$ (ppm) | $NH_3$ (ppb) | $NO_2$ (ppb) | RH (%) | T (°C) | SOA mass conc. ($\mu g\ m^{-3}$) |
|---|---|---|---|---|---|---|---|
| Exp.1 | 790 | 1.98 | - | - | 25±1 | 20±1 | 637±14.6 |
| Exp.2 | 790 | 1.98 | 200 | - | 23±1 | 20±1 | 867±12.7 |
| Exp.3 | 790 | 1.98 | 200 | 62 | 26±1 | 20±1 | 1020±10.6 |
| Exp.4 | 790 | 1.98 | - | 63 | 25±1 | 20±1 | 452±18.9 |

## Figures

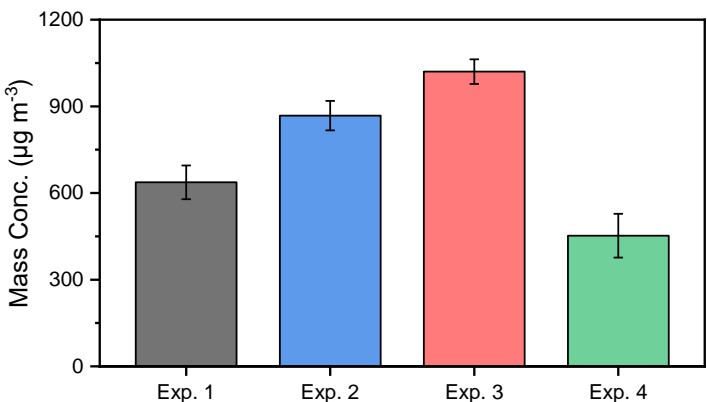

**Fig. 1.** Maximum mass concentration of toluene-derived SOA in different experiments. All the
mass concentrations were wall-loss corrected.





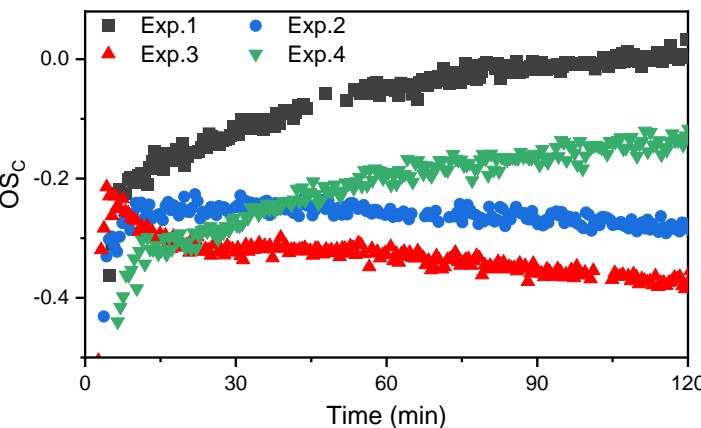


**Fig. 2.** The $OS_C$ values for the toluene SOA formed under different $NH_3$/NOx conditions.



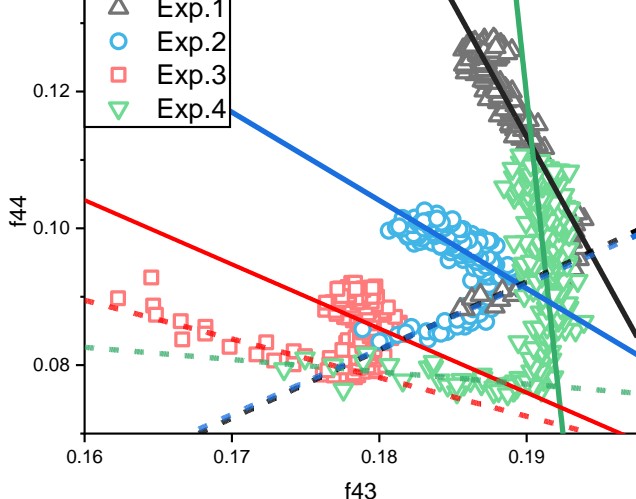


**Fig. 3.** The relationship between total organic signals at 43 m/z (f43) *vs.* 44 m/z (f44) from SOA
data during the photooxidation process. The f43 *vs.* f44 plots exhibited inflection points during the
photooxidation process. The dashed lines indicate the trends of f43 *vs.* f44 for the SOA formation
stage (before the inflection point) and the solid lines for the stable stage.



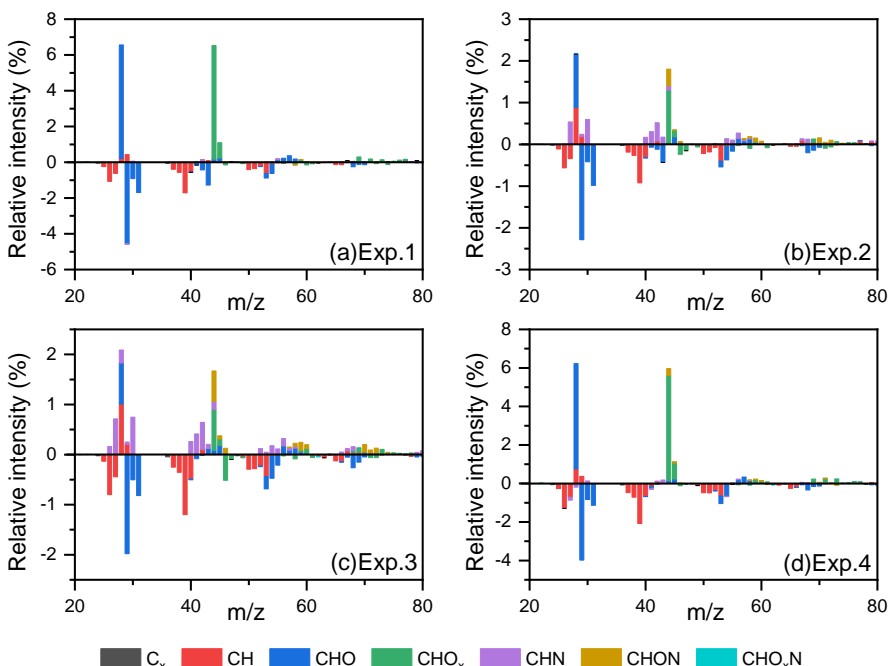


**Fig. 4.** The differential spectra of toluene SOA in the formation and stable stages. Data were taken
and analyzed at a high resolution but were summarized to a unit mass resolution for display.



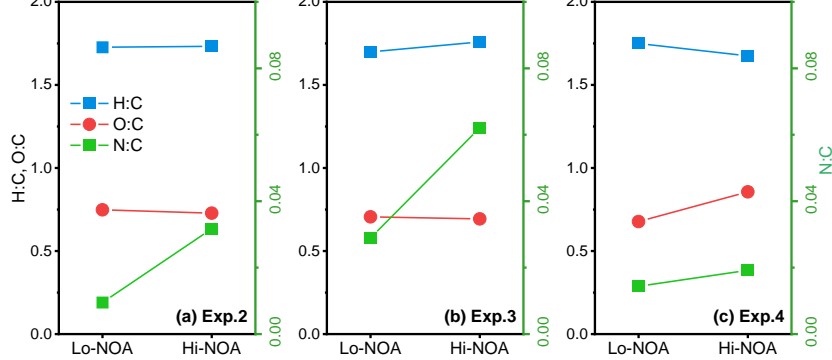


**Fig. 5.** The H/C, O/C, and N/C values of Hi-NOA and Lo-NOA for each experiment. (a) Exp. 2
with 200 ppb $NH_3$, (b) Exp. 3 with 200 ppb $NH_3$ and 62 ppb $NO_2$, and (c) Exp.4 with 63 ppb $NO_2$.


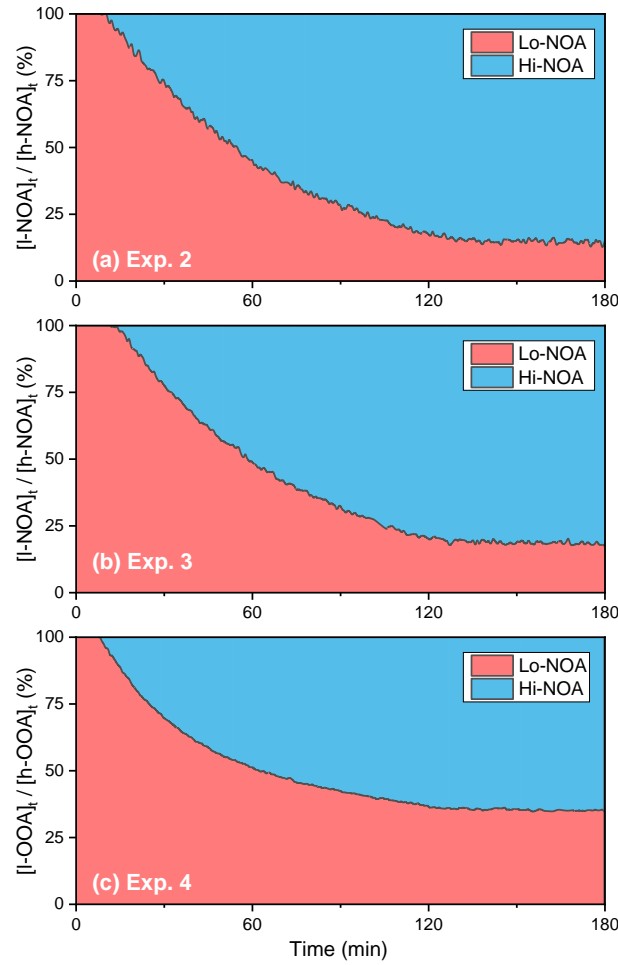


**Fig. 6.** The evolution of high-nitrogen OA (Hi-NOA) and low-nitrogen OA (Lo-NOA) during the
photooxidation process under different NOx/NH$_3$ concentrations. Hi-NOA and Lo-NOA were not
consistent among experiments. (a) Exp. 2 with 200 ppb NH$_3$, (b) Exp. 3 with 200 ppb NH$_3$ and 62
ppb NO$_2$, and (c) Exp. 4 with 63 ppb NO$_2$.




**Fig. 7.** The evolution of N/C in different experiments.





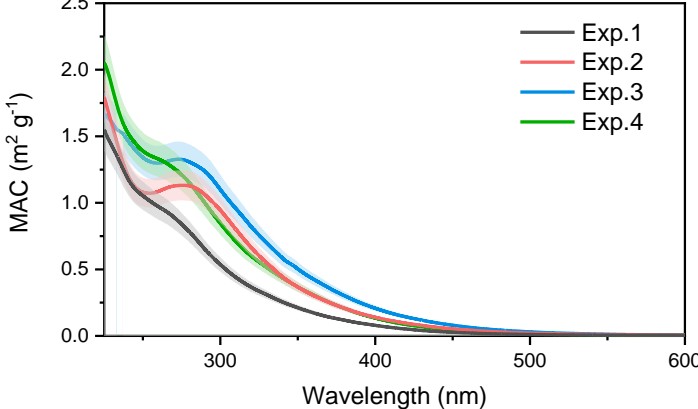


**Fig. 8.** The MAC over the range of 200–600 nm for the toluene SOA formed under different
experiment conditions.