# Peer review of "Synergetic effects of NH3 and NOX on the production"

_Atmospheric Chemistry and Physics, 2021_

## Referee Comment (RC1)

The script by Liu et al. reported synergetic effects of $NH_3$ and NOx on toluene SOA production, chemical composition, and optical properties. Though experiment was not well designed nor some important information (experiment repetition, photon-flux, OH radical exposure, details in wall-loss correction, etc.) on chamber operation was provided, their results are interesting and important if all their speculations on chemical results were reliable based on simple AMS measurement. More chemical information on molecular levels and proper reaction pathways should be provided to verify the discussion. Besides, the authors should go through script carefully to check all the values and units. Overall, the decision is Major revision and the authors should address all the comments.

Specific comments:

1. Line 107: "participate"

2. Line 148: Particle wall loss rate is constrained by many factors, including particle size, chemical composition, and wall environment. You can surely use standard aerosols to derive a simple wall loss rate for ensemble particles in the chamber, but detailed information concerning the loss-rate should be provided and also discuss the feasibility to apply one rate to correct all experiments with different environments.

3. In method description, what was the shape of your chamber? how do you control the chamber temperature when the whole teflon bag in operation was sealed in light box? And what is the light transition efficiency of your teflon bag? How to make sure the chemicals were eventually distributed in the chamber?

4. What was the general OH radical concentration throughout each experiment? Did you quantify the OH radical exposure to ambient environment? Was it atmospheric environmental relevant for toluene photooxidation?

5. Line 163: "NOx (……) or/and $NH_3$"

6. Line 169: delete "present"

7. Line 176: SMPS classifies electrical mobility size of particles.

8. Line 180: Toluene SOA density shall be variant in response to different ageing degree and presence of NOx or NH$_3$.

9. Line 203: commonly W-mode was suggested to derive elemental ratio of organics from AMS results.

10. Line 214: is "46.2 mm PTFE filter"?

11. Line 215: what was the extraction efficiency? In presence of NOx/NH$_3$, inorganics shall form along with toluene SOA. Did you consider the salt influence on the extract absorption? and also interference on AMS characterization of organic fractions? It should be noted that ammonium nitrate has interference on AMS ionization, f44 signal and associated elemental ratios.

12. Line 229: how did you quantify methanol-soluble organic carbon in clear methanol solution? if MAC was calculated on the basis of organic carbon, the unit of M should be $\mu$g OC m$^{-3}$ .

13. In experiments, did you feed air to the chamber during monitoring and sampling? If so, did you consider the dilution effect on your final results on SOA concentration? How many times you repeated each test?

14. Line 233: change "during" to "on"

15. Line 255: change to "this may at least partly explain……"

16. Line 266-267: confusing, make the statement clear

17. Line 275: in current toluene photooxidation with NOx and NH$_3$ presence, nitrogen content is significant enough to be considered in carbon oxidation state calculation. The simplified OSc may be biased in chemical feature description of toluene SOA.

18. Line 291: delete "toward", change to "continuing increase of SOA and its OSc"

19. Line 284-300: time-profile of SOA mass concentration from UV on till end of experiment was needed to support your speculation of OSc changes. Besides, Organic ($C_xH_y^+$, $C_xH_yO^+$, $C_xH_yO_i^+$, etc.) and inorganic contribution over the course of each test should be provided. Can you tell at what proper time OH radical was totally consumed in the chamber since UV on?

20. Line 308: confusing. Ammonium salt offsets what?

21. Why not use practical m/z-43 ($C_2H_3O^+$) and m/z-44 ($CO_2^+$) mass concentration changes to describe proper reaction pathways in toluene photooxidation? Mass fraction can only tell relative change.

22. In tracking Hi-NOA and LO-NOA via PMF, have you ever considered application of some characteristic fragments indicating products by homogeneous acid-base reactions and heterogeneous Maillard reactions to verify your hypothesis? Molecular analysis by HRMS is most reliable compared to AMS redrived results.

23. Line 439: why dose Lo-NOA have higher N/C ratio? 0.062 *vs*. 0.29 for Ho-NOA *vs*. Lo-NOA?

24. Line 449: how did you count the values of N/C ratio and percentage? Check all the values in the script carefully. 0.0404 and 0.216 accounts for 65% and 35 %, respectively?

25. SOA concentrations have been summarized in Table 1, Figure 1 is not necessary, remove it. Besides, it is suggested to add photon flux and OH radical concentration information in Table 1.

26.  Redraw Figure 3, make it clearer. Suggest to add time information to trace time-profile of *f*43-*f*44.

27. Make clear statement of specific time for formation and stable stage of SOA in Figure 4, is it stable minus initial stage? Explain the presence of nitrogen-bearing fragments in Experiment 1.

28. Present error and statistic results in Figure 5.

29. Keep consistent of axis label and legend in Figure 6.

---

## Author Response (AR1)

**Response to Reviewers**

Ms. Ref. No.: acp-2021-560

Synergetic effect of $NH_3$ and NOx on the production and optical absorption of secondary organic aerosol formation from toluene photooxidation

Dear Prof. Mu:

    We greatly appreciate your comments on our manuscript. All the comments are very helpful to improve the quality of our manuscript. Our responses to your specific comments/questions are itemized below.

    All the best

Gehui Wang

October 14, 2021

The authors of this manuscript investigated the influence of $NH_3$ and NOx on SOA formation from photooxidation of toluene. They concluded that there was a synergistic effect of NH3 and NOx on toluene-SOA formation based on the evident increase of SOA concentration as well as the mass absorption coefficient for the photooxidation system of toluene +H2O2+ NH3 +NOx in comparison with those of toluene +H2O2, toluene +H2O2 +NH3 and toluene +H2O2 + NOx. If the conclusion was reliable, it would be of great importance for evaluating atmospheric SOA formation from photooxidation of VOCs. The manuscript is recommended to be published in the journal after considering the following aspects:

1. It is better to use SOA yields, rather than SOA mass concentrations, to elucidate the influence of $NH_3$ and NOx on SOA formation from photooxidation of toluene, because the consumption of toluene was different for the four experiments especially for the irradiated mixtures with the presence of NOx which would significantly suppress the

OH level in the photochemical chamber. Did you measure the concentration variation of toluene during the experiments?

**Author reply:**

We are very grateful to Prof. Mu for this comment. We also think the toluene consumption is very necessary for the SOA formation study. We measured the toluene concentration with PRT-MS on-line. The evolution of toluene concentration at different experiment conditions was shown below. The evolution of toluene concentration was almost uniformly with and without NOx. NOx could compete with toluene to react with OH in the chamber. Because the NOx concentration is much lower than that of toluene as shown in Table 1. The effect of NOx on the consumption of toluene was not significant in this study.

[Figure]

Fig.S1 The evolution of toluene concentration for each experiment.

In the revised manuscript, we have added the sentences in line 163 as follows:

"Toluene concentration was measured with a Proton Transfer Reaction-Mass Spectrometry (PTR-ToF-MS, Ionicon Analytik, Austria). The evolution of toluene concentration for different experiments was shown in Fig.S1."

we also added the sentences in line 235 as follows:

The sentence of "The SOA mass concentrations at different conditions are shown in Fig. 1." in line 234 was changed as "SOA yield (Y) is defined as Y = $\Delta M_0/\Delta HC$, where $\Delta M_0$ is the produced organic aerosol mass concentration ($\mu g\ m^{-3}$), and $\Delta HC$ is the mass concentration of reacted toluene ($\mu g\ m^{-3}$). The evolution of SOA mass concentrations and SOA yield at different conditions during the photooxidation process were shown in Fig.1."

The Fig. 1 in line 798 was updated as below:

[Figure]

Fig. 1. The evolution of mass concentration (a) and yield (b) of toluene-derived SOA in different experiments. All the mass concentrations were wall-loss corrected. The error bars of SOA yield were calculated by the volatility of measured SOA concentration after the UV light was turn off at the end of each experiment.

Table 1 was also updated as below:

**Table 1.** Summary of experimental conditions in this study

| No. | $Tol_0$ (ppb) | $\Delta$ Tol (ppb) | $NH_3$[a] (ppb) | $NO_2$ (ppb) | RH (%) | SOA mass conc.[b,c] ($\mu g\ m^{-3}$) | SOA yield[b] (%) |
|---|---|---|---|---|---|---|---|
| Exp.1 | 664.1 | 551.2 | - | - | 25 ± 1 | 637 ± 14.6 | 28.1 |
| Exp.2 | 618.7 | 499.4 | ~200 | - | 23 ± 1 | 867 ± 12.7 | 34.7 |
| Exp.3 | 620.9 | 526.1 | ~200 | 62 | 26 ± 1 | 1020 ± 10.6 | 42.7 |
| Exp.4 | 645.7 | 539.5 | - | 63 | 25 ± 1 | 452 ± 18.9 | 19.5 |

[a] The concentration of $NH_3$ is estimated by the amount of $NH_3$ added and the volume of the smog chamber. [b] SOA concentration and yield was calculated after taking wall loss into account. [c] The reported SOA mass concentrations was peak values after the wall loss correction.

2. It is better to present the variation trend of particle mass concentration measured by the SMPS for each experiment because the readers will not understand the detail information about the maximal mass concentration in Fig. 1. Did you conduct duplicate or triplicate experiments for each case? This information should mention in the experimental section, or readers cannot understand how the error bars in Fig. 1 come from.

**Author reply:**

The variation trend of particle mass concentration measured by the SMPS for each experiment was shown below and we added it into the manuscript as described in the reply of Referee #1. We changed Fig.1 as below:

[Figure]

**Fig. 1**  The evolution of mass concentration (a) and yield (b) of toluene-derived SOA in different experiments. All the mass concentrations were wall-loss corrected. The error bars of SOA yield were calculated by the volatility of measured SOA concentration after the UV light was turn off at the end of each experiment."

The repeat experiments were not carried out here. The error bars in Fig. 1 were calculated by the fluctuation of SOA concentration when the UV light was turn off at the end of each photooxidation experiment. In order to make this section clearer, the sentence of "The error bars were calculated by the volatility of measured SOA concentration after the UV light was turn off at the end of each experiment." was added in the caption of Fig. 1.

3. Particle mass concentration measured by the SMPS couldn't be simply attributed to SOA because evident amount of secondary inorganic aerosol (such as ammonium and nitrate) could be formed in the photochemical reaction systems with

the presence of $NH_3$ and NOx. If $NH_3$ was totally converted into particulate ammonium, there would be no evident influence of $NH_3$ on toluene-SOA. As the reaction of $NO_2$ with OH radical is very fast, most of $NO_2$ will be quickly converted into $HNO_3$ and $NH_4NO_3$ (with the $NH_3$ presence), which may act as a seed to accelerate toluene-SOA formation. Did you measure the concentration variations of $NH_3$, NOx, $NO_3^-$ and $NH_4^+$ during the experiments? This information is valuable for explaining the experimental results.

**Author reply:**

The concentration variation of NOx was measured during the experiments, the $NH_3$ concentration was not measured here. The concentration variations of $NO_3^-$ and $NH_4^+$ during the experiments were measured by the AMS. The content of $NO_3^-$ and $NH_4^+$ is very low in particulate matter. The content of $NO_3^-$ and $NH_4^+$ for each experiment was list in the table below.

**Table S1.** The content of $NO_3^-$ and $NH_4^+$ in the particle-phase for each experiment

|  | $[NOx]_0$ (ppb) | $[NH_3]_0$ (ppb) | $[NO_3^-]/[Org]$ (%) | $[NH_4^+]/[Org]$ (%) |
|---|---|---|---|---|
| Exp.2 | - | ~200 | - | 1.9 |
| Exp.3 | 62 | ~200 | 4.0 | 2.6 |
| Exp.4 | 63 | - | <0.2 | - |

If only $NO_2$ was introduced in the chamber, the content of $NO_3^-$ in particulate matter was less than 0.2%. Most of the $NO_3^-$ was in the gas phase in the form of nitric acid. If only $NH_3$ was introduced in the chamber, the content of $NH_4^+$ was about 1.9%. when both $NO_2$ and $NH_3$ were introduced in the chamber, the $NO_3^-$ reacted with $NH_4^+$ to form ammonium nitrate particles. Large amounts of $NO_3^-$ was participated into the particle phase nitrate. The inorganic matter could account for 6.6% of the total mass of particulate matter. However, the low content of inorganic matter still not affect the results of the synergetic effect of $NH_3$ and NOx on the toluene SOA formation. The following sentences were added in line 253 of the manuscript to explain the smaller

effect of the inorganic aerosol on the enhancement in particle mass in the $NH_3 + NOx$ experiment.

"Although inorganic aerosol was formed from the interaction of $NH_3$ and NOx in the chamber, the inorganic matter only accounted for 6.6% of the total mass of particulate matter (Table S1) in the $NH_3 + NOx$ experiment. Thus, it was not the main cause of the increase in particulate matter."

4. The exact concentrations of the reactants listed in Table 1 are unbelievable, it is better replaced by "~value".

**Author reply:**

We have revised it, and the new Table 1 has been shown in Comment #1.

5. The concentration of toluene used for the experiments is more than two orders of magnitude than those observed in polluted areas, and thus the results may be less representative for the actual atmosphere and exaggerate the influence of $NH_3$ on toluene-SOA because of significant formation of carbonyl and carboxyl compounds. The authors are suggested to conduct experiments by using lower concentrations of toluene (e.g., ~50-100ppb) to check the possible dependence of the influence on the initial concentration of toluene.

**Author reply:**

We fully agree with the reviewer on this opinion. Considering the performance of this chamber, as well as to obtain more accurate analysis of chemical composition, a higher toluene was used in this study. In our work, the proportion of the key components is comparable to the ambient level. The mean concentration of aromatics VOCs and OH was about 11 ppb (Zou et al., 2015) and $1 \times 10^6$ molecule cm$^{-3}$ (Prinn et al., 1995), respectively, which was consist with the VOCs/OH ratio (~700 ppb/$5.9 \times 10^7$ molecule cm$^{-3}$) in our study.

To provide a better illustration of the experiment condition, the following sentences have been added in the revised manuscript in line 170:

"In our work, the OH and toluene concentrations were higher than those of urban conditions. The purpose of the high OH and toluene concentrations is to obtain enough particle production samples for off-line collections and accurate measurements. The toluene concentrations remained stable under the different experimental conditions, the variation of toluene-derived SOA mass concentration and yield was only affected by the different $NO_2$ and/or $NH_3$ concentrations in this study. Toluene was studied here as the representative of total aromatic VOCs in the urban atmosphere. The concentration ratio of toluene to OH in this study is similar to that under the real atmospheric conditions (Zou et al., 2015; Prinn et al., 1995)."

In the revised manuscript, we have added the sentences in line 521 as follows:

"It has to be noted that the concentration of reactants used for the experiments is much higher than that observed in polluted areas, the effect of $NH_3$ and NOx on the photooxidation of toluene with lower concentrations would be checked in the further study."

6. SOA formation is usually affected by OH levels, could you estimate the OH levels based on the first order decay of toluene?

**Author reply:**

The highest OH concentration of $1.02 \times 10^8$ molecule cm$^{-3}$ was observed at the beginning of the reaction. The average OH concentration over the entire reaction period was $5.87 \times 10^7$ molecule cm$^{-3}$.

In the revised manuscript, we have added the sentences in line 163 as follows:

"The OH concentration in the chamber was calculated based on the first order decay of toluene concentration. There was no obvious difference of OH concentrations in the different NOx and $NH_3$ conditions (Fig. S2)."

The calculation of OH concentration was added in the Supporting Information as below:

"**S1 OH Concentration Calculation Process**

The OH concentration was calculated based on the decay ratio of toluene concentrations and the known rate constant with respect to OH. The change of toluene concentration over time can be expressed as:

$$-\frac{d[toluene]}{dt} = K_{OH} \times [OH] \times [toluene] \tag{RS1}$$

Where, $K_{OH}$ is the reaction rates constant of OH radicals with toluene ($K_{OH}=5.7\times10^{-12}$ $cm^3$ $molecule^{-1}$ $s^{-1}$). Assuming that the concentration of hydroxide did not change during the experiment, then we can get:

$$\ln(\frac{[toluene]_0}{[toluene]_t})/t = K_{OH} \times [OH] \tag{RS2}$$

Thus, plotting the variation curve of $\ln([toluene]_0/[toluene]_t)$ vs. time t showed as Fig.S1. The $\ln([toluene]_0/[toluene]_t)$ in Fig.1(b) was not a straight line. This is because the OH is consumed as the reaction goes on. The evolution of OH concentration at experiment conditions was shown in Fig.S2. The different experiment conditions in this study did not affect the OH concentration obviously. The highest OH concentration of $1.02 \times 10^8$ molecule $cm^{-3}$ was observed at the beginning of the reaction. The average OH concentration over the entire reaction period is $5.87 \times 10^7$ molecule $cm^{-3}$.

[Figure]

Fig.S2 The evolution of OH concentrations at different experiment

conditions"

**Reference**

Prinn, R. G., Weiss, R. F., Miller, B. R., Huang, J., Alyea, F. N., Cunnold, D. M., Fraser, P. J., Hartley, D. E., and Simmonds, P. G.: Atmospheric trends and lifetime of $CH_3CCl_3$ and global OH concentrations, Science, 269, 187-192, 10.1126/science.269.5221.187, 1995.

Zou, Y., Deng, X. J., Zhu, D., Gong, D. C., Wang, H., Li, F., Tan, H. B., Deng, T., Mai, B. R., Liu, X. T., and Wang, B. G.: Characteristics of 1 year of observational data of VOCs, NOx and $O_3$ at a suburban site in Guangzhou, China, Atmos. Chem. Phys., 15, 6625-6636, 10.5194/acp-15-6625-2015, 2015.

**Response to Reviewers**

Ms. Ref. No.: acp-2021-560

Synergetic effect of NH$_3$ and NOx on the production and optical absorption of secondary organic aerosol formation from toluene photooxidation

Dear Editor:

We greatly appreciate the time and effort that the editor and reviewer spent in reviewing our manuscript. After reading the comments from the reviewers, we have carefully revised our manuscript. All the changes we made are marked in red. Our responses to the comments are itemized below. The referee's comments are in black, authors' responses are in blue.

Anything for our paper, please feel free to contact me via **ghwang@geo.ecnu.edu.cn.**

All the best

Gehui Wang

Oct. 14, 2021

**Reviewer #1**

The script by Liu et al. reported synergetic effects of NH$_3$ and NOx on toluene SOA production, chemical composition, and optical properties. Though experiment was not well designed nor some important information (experiment repetition, photon-flux, OH radical exposure, details in wall-loss correction, etc.) on chamber operation was provided, their results are interesting and important if all their speculations on chemical results were reliable based on simple AMS measurement. More chemical information on molecular levels and proper reaction pathways should be provided to verify the discussion. Besides, the authors should go through script carefully to check all the values and units. Overall, the decision is Major revision and the authors should address all the comments.

**Author reply:**

We thank the reviewer for the important comments, which are very helpful for improving our paper quality. We have carefully revised the manuscript, see the details below.

Specific comments:

1. Line 107: "participate"

**Author reply:**

"participated" is fixed as "partition".

2. Line 148: Particle wall loss rate is constrained by many factors, including particle size, chemical composition, and wall environment. You can surely use standard aerosols to derive a simple wall loss rate for ensemble particles in the chamber, but detailed information concerning the loss-rate should be provided and also discuss the feasibility to apply one rate to correct all experiments with different environments.

**Author reply:**

We agree with the Referee that the wall loss rates are is constrained by many factors. In this study all the particle mass concentration was corrected by using the method same as that of Jiang et al. (2020) and Pathak et al. (2007) to constrain the influence of wall losses of different SOA formed with different experiment conditions. For each experiment, we continued to monitor the particle concentration in the dark condition for 1 hour, and recalculated the particle wall loss constant according to the variation of particle concentration. After the wall loss correction, we found that he particle mass concentration was almost constant (New Fig.1), thus we believe that our results are reliable and credible.

To clarify the statement, we added the sentences in line 235 of the revised manuscript: "However, the particle wall loss rates were detected at the end of the chamber experiment after the UV-lamps were turned off, and the mass concentration was corrected with the method reported by Jiang et al. (2020) and Pathak et al. (2007). After the wall loss correction, the particle mass concentration was almost constant, the

different wall loss effect caused by gaseous oxidation products formed under the different experiment conditions have been remedied."

3. In method description, what was the shape of your chamber? how do you control the chamber temperature when the whole teflon bag in operation was sealed in light box? And what is the light transition efficiency of your teflon bag? How to make sure the chemicals were eventually distributed in the chamber?

**Author reply:**

The shape of the chamber is a cube-like.

The temperature of the chamber located room was controlled at 20 °C by two air conditioners. Two fans for air exchange inside and outside the chamber room to ensure the stable temperature of the chamber. However, the temperature in the chamber still increased when the UV light turn on. The chamber temperature increased from to 25 °C within 30 min, then the temperature become stable. The evolution of the temperature in chamber was consistent for each experiment, and not the environment factors that affect the SOA formation in this study.

The sentence of "All experiments were performed at room temperature (293~298 K) and one atmospheric pressure was maintained in the chamber at all time." was added in line 153.

Light transition efficiency of this Teflon bag is not known here, but we can make sure light intensity in the chamber is consistent for each experiment.

VOCs, $H_2O_2$, NOx and $NH_3$ were added into the chamber with the zero air. The flow of zero air is 20 L $min^{-1}$. When zero air is added, it has a good stirring effect on the gases in the chamber. The reactants in the chamber can be evenly mixed within 2 min. To ensure that VOC is already evenly distributed in the chamber, we let the chamber stand for about 10 minutes. Meanwhile, we measured the concentration of VOC by the PTR-MS and found it was not changed. Hence, we believe that 10 min after the reactants were added, chemicals in the chamber were well mixed.

This sentence was added in line 169: "After all the reactants were added, the chamber stood quietly for 10 min without turning on the light to ensure that the reactant gases in the chamber were evenly mixed."

4. What was the general OH radical concentration throughout each experiment? Did you quantify the OH radical exposure to ambient environment? Was it atmospheric environmental relevant for toluene photooxidation?

**Author reply:**

We are very grateful to the reviewer for this comment.

In the revised manuscript, we have added the calculation of OH concentrations based on the first order decay of toluene in line 163 as follows: "The OH concentration in the chamber was calculated based on the first order decay of toluene concentration. There was no obvious difference of OH concentrations in the different NOx and NH$_3$ levels (Fig. S2)."

The highest OH concentration of $1.02 \times 10^8$ molecule cm$^{-3}$ was observed at the beginning of the reaction. The average OH concentration over the entire reaction period was $5.87 \times 10^7$ molecule cm$^{-3}$. The details on the calculation of OH concentration was added in the Supporting Information as below:

"**S1 OH Concentration Calculation Process**

The OH concentration was calculated based on the decay ratio of toluene concentrations and the known rate constant with respect to OH. The change of toluene concentration over time can be expressed as:

$$-\frac{d[\text{toluene}]}{dt} = K_{\text{OH}} \times [\text{OH}] \times [\text{toluene}] \tag{RS1}$$

Where, $K_{\text{OH}}$ is the reaction rates constant of OH radicals with toluene ($K_{\text{OH}}=5.7 \times 10^{-12}$ cm$^3$ molecule$^{-1}$ s$^{-1}$). Assuming that the concentration of hydroxide did not change during the experiment, then we can get:

$$\ln(\frac{[\text{toluene}]_0}{[\text{toluene}]_t})/t = K_{\text{OH}} \times [\text{OH}] \tag{RS2}$$

Thus, plotting the variation curve of $\ln([\text{toluene}]_0/[\text{toluene}]_t)$ vs. time t showed as Fig.S1. The $\ln([\text{toluene}]_0/[\text{toluene}]_t)$ in Fig.1(b) was not a straight line. This is because the OH is consumed as the reaction goes on. The evolution of OH concentration at experiment conditions was shown in Fig.S2. The different experiment conditions in this study did not affect the OH concentration obviously. The highest OH concentration of

1.0 $\times$ $10^8$ molecule cm$^{-3}$ was observed at the beginning of the reaction. The average OH concentration over the entire reaction period is 5.9 $\times$ $10^7$ molecule cm$^{-3}$.

[Figure]

Fig.S2 The evolution of OH concentrations at different experiment conditions"

The atmosphere is very complicated, and the experimental conditions in the chamber cannot be the same as the real environment. In our work, the proportion of the key components is comparable to the real atmosphere. The mean concentration of aromatics VOCs and OH was about 11 ppb (Zou et al., 2015) and 1 $\times 10^6$ molecule cm$^{-3}$ (Prinn et al., 1995), respectively, which was consist with the VOCs/OH ratio (~700 ppb/5.9 $\times 10^7$ molecule cm$^{-3}$) in our study.

To provide a better illustration on the experiment condition, the following sentences have been added into the revised manuscript in line 170:

"In our work, the OH and toluene concentrations were higher than those of urban conditions. The purpose of the high OH and toluene concentrations is to obtain enough particle production samples for off-line collections and accurate measurements. The toluene concentrations remained stable under the different experimental conditions, the variation of toluene-derived SOA mass concentration and yield was only affected by the different NO$_2$ and/or NH$_3$ concentrations in this study. Toluene was studied here as the representative of total aromatic VOCs in the urban atmosphere. The concentration ratio of toluene to OH in this study is similar to that under the real atmospheric conditions (Zou et al., 2015; Prinn et al., 1995)."

**Author reply:**

Fixed.

6. Line 169: delete "present"

**Author reply:**

Deleted.

7. Line 176: SMPS classifies electrical mobility size of particles.

**Author reply:**

We added "(from 14.1 nm to 736.5 nm)" after "which were used for screening particles with specific aerodynamic equivalent sizes" in line 176.

8.Line 180: Toluene SOA density shall be variant in response to different ageing degree and presence of NOx or NH$_3$.

**Author reply:**

We agree with the reviewer on this comment. Toluene SOA density can be variant with different experiment condition and oxidation time. It is indisputable that if we got real-time density of SOA, it will further improve the accuracy of our result. A density of 1.3 g m$^{-3}$ has been widely used for toluene SOA in the previous studies (Volkamer et al., 2006; Qi et al., 2020; Ng et al., 2007; Liu et al., 2021; Ji et al., 2017). We also believe that the density used here is reliable, and the variation of SOA density in different ageing degree and experiment conditions will not significantly affect our results.

9. Line 203: commonly W-mode was suggested to derive elemental ratio of organics from AMS results.

**Author reply:**

We thank for this comment. V-mode is able to meet the requirement of our experiment. We did not use W-mode in this study.

10. Line 214: is "46.2 mm PTFE filter"?

**Author reply:**

Yes. The label was shown below.

[Figure]

11. Line 215: what was the extraction efficiency? In presence of NOx/NH₃, inorganics shall form along with toluene SOA. Did you consider the salt influence on the extract absorption? and ammonium nitrate also interference on AMS characterization of organic fractions? It should be noted that ammonium nitrate has interference on AMS ionization, f44 signal and associated elemental ratios.

**Author reply:**

Chen and Bond (2010) reported that > 92 % of SOA was extractable by organic solvents (methanol or acetone). The uncertainty is probably insignificant in this study. We add the follow sentence in line 215: "As reported by Chen and Bond (2010), > 92 % of SOA is extractable by organic solvents (e.g., methanol), which means that almost all organic matter was extracted in this study."

We think the salt influence is insignificant, because the amount of methanol is 5 mL, which is much larger than the SOA mass.

AMS is widely used to quantify OA elemental composition, oxidation state in the field observations. Pieber et al. (2016) show that a non-OA $CO_2^+$ signal can arise from reactions on the particle vaporizer, ion chamber, or both, induced by thermal decomposition products of inorganic salts. They have pointed $NH_4NO_3$ could cause a median $CO_2^+$ interference signal of +3.4% relative to nitrate. Although, the propagation of the $CO_2^+$ interference to other ions during standard AMS data analysis affects the calculated OA mass, mass spectra, molecular oxygen-to-carbon ratio (O/C), and f44. But the resulting bias may be trivial for most ambient data sets but can be significant for aerosol with higher inorganic fractions (>50%) (Pieber et al., 2016). In this study, the proportion of $NH_4NO_3$ in the particle-phase is lower than 7%. The interference of $NH_4NO_3$ on AMS ionization, f44 is not significant. We believe our AMS data is credible here.

12. Line 229: how did you quantify methanol-soluble organic carbon in clear methanol solution? if MAC was calculated on the basis of organic carbon, the unit of M should be µg OC m$^{-3}$.

**Author reply:**

The MAC was calculated on the basis of total particulate matter, not organic carbon. The collected particle mass was measured by weighing the weight of the Teflon filter before and after the SOA collection. Unit of M (µg m$^{-3}$) used here was corrected.

13. In experiments, did you feed air to the chamber during monitoring and sampling? If so, did you consider the dilution effect on your final results on SOA concentration? How many times you repeated each test?

**Author reply:**

The chamber volume is variable in this study, thus we did not feed air during the experiment period. The total sample volume for the online measurement was only 3 L min$^{-1}$. The illumination time of UV light was 120 min. The total sample volume was lower than 8% of the total volume. The change rate of chamber volume is even lower than the systematic error of SMPS (10%) (Liu et al., 2017a). Therefore, the change of

chamber volume did not significantly affect the photooxidation process and SOA formation for each experiment. The experiment was not repeated in this study.

14. Line 233: change "during" to "on"

**Author reply:**

Suggestion taken.

15. Line 255: change to "this may at least partly explain……"

**Author reply:**

Suggestion taken.

16. Line 266-267: confusing, make the statement clear

**Author reply:**

This sentence was revised as "resulting in the transformation of SOA chemical composition continuously proceeding during the photooxidation process, but little attention has been paid to the evolution of SOA chemical composition in previous studies."

17. Line 275: in current toluene photooxidation with NOx and NH₃ presence, nitrogen content is significant enough to be considered in carbon oxidation state calculation. The simplified OSc may be biased in chemical feature description of toluene SOA.

**Author reply:**

We express our appreciation to the reviewers for this comment.

We added the N/C in the calculation of $OS_C$. When SOA generated under $NH_3$ condition in Exp.2, the particulate nitrogen is fully reduced ($OS_N = -3$), so $OS_C$ was determined by $OS_C = 2\,O/C - H/C + 3\,N/C$. When SOA generated under NOx condition in Exp.4, the particulate nitrogen is almost certainly in the form of organic nitrates ($OS_N$

= +5), so $OS_C$ was determined by $OS_C = 2$ O/C - H/C - 5 N/C. New $OS_C$ figure was shown below:

[Figure]

The new $OS_C$ value still ranging between -0.5 and 0. The evolution of $OS_C$ for different experiments was no change. And it does not make any difference to our conclusions.

The sentence "$OS_C$ is calculated based on the measurements of O/C and H/C ($OS_C \approx 2 \times$ O:C - H:C)." in line 274 was revised as "Average $OS_C$ calculation was shown in Supporting Information."

The following sentence was added in SI:

"**S2 $OS_C$ calculation**

In most previous studies, $OS_C$ was estimated from the O/C and H/C data (Liu et al., 2015; Chen et al., 2019; Chhabra et al., 2011; Docherty et al., 2018; Kroll et al., 2011). Because nitrogen content is significant enough to be considered in carbon oxidation state calculation of Exp.2 - 4 in our study, the $OS_C$ value used here was calculated based on the O/C, H/C and N/C ratio.

For Exp.2 with $NH_3$ presence, particulate nitrogen is almost certainly in the form of ammonium salt with nominal oxidation numbers of -3. Alternatively, average $OS_C$ in Exp. 2 was calculated using the equation of $OS_C = 2$ O/C – H/C +3 N/C. For Exp.4, toluene SOA formed with NOx presence, particulate nitrogen is almost certainly in the form of organic nitrates (i.e., $-ONO_2$ with nominal oxidation numbers of +5) (Park et al., 2017; Ruggeri et al., 2016). Alternatively, average $OS_C$ in Exp. 4 can also be simply

calculated using the following equation of intensity-weighted mean O/C, H/C, and N/C: $OS_C = 2\ O/C - H/C - 5\ N/C$.

Both NH$_3$ and NOx was presence in Exp.3, we estimated the contribution of NH$_3$ and NOx to organic nitrogen based on N:C value in Exp. 2 and Exp. 4, respectively. Average $OS_C$ in Exp. 3 was calculated as: $OS_C = 2\ O/C - H/C + \sigma_{NH3} \times 3\ N/C - \sigma_{NOx} \times 5\ N/C$. Here, $\sigma_{NH3}$ is the contribution rate of NH$_3$ to total organic nitrogen in SOA, and $\sigma_{NOx}$ is the contribution rate of NOx to total organic nitrogen in SOA."

18. Line 291: delete "toward", change to "continuing increase of SOA and its OSc"

**Author reply:**

Suggestion taken.

19. Line 284-300: time-profile of SOA mass concentration from UV on till end of experiment was needed to support your speculation of OSc changes. Besides, Organic (C$_x$H$_y^+$, C$_x$H$_y$O$^+$, C$_x$H$_y$O$_i^+$, etc.) and inorganic contribution over the course of each test should be provided. Can you tell at what proper time OH radical was totally consumed in the chamber since UV on?

**Author reply:**

The contribution of CH, CHO and CHOx groups over the course of each test was provided in SI.

[Figure]

Fig. S3 The evolution of OH concentrations at different experiment conditions"

The following sentence was added in Line 283: "The evolution of CH, CHO and CHOx proportions to total organic matter was shown in Fig. S3. In Exp. 1 and 4, CHOx proportion increased with photooxidation time, the contribution of more highly oxidized products to SOA was gradually increased, which was consist with the increasing $OS_C$ value."

The real OH concentration was shown before. About 90 min after the UV light was turned on, OH concentration is almost zero. Our objective here is to illustrate the possibility of the influence of UV light on SOA chemical composition. To make it clear, the sentences in line 295-300 of the manuscript were revised as: "Finally, as pointed by Malecha and Nizkorodov (2016), even if there was no OH radical in the chamber, the photodegradation of SOA can produce small oxygenated volatile organic compounds (e.g. acetaldehyde $OS_C$=-1, and acetone $OS_C$≈-1.3) under UV light irradiation. The photoproduction of OVOCs from SOA had a lower $OS_C$ value than that of SOA.

Although the loss of SOA through photodegradation is small, the $OS_C$ value of SOA still had increased to a certain extent (Malecha and Nizkorodov, 2016)."

20. Line 308: confusing. Ammonium salt offsets what?

**Author reply:**

This sentence was revised as "An organic ammonium salt with four H atoms can offset an increase in $OS_C$ value caused by the formation of organic acids/carboxyl group with two O atoms through acid-base reactions."

21. Why not use practical m/z-43 ($C_2H_3O^+$) and m/z-44 ($CO_2^+$) mass concentration changes to describe proper reaction pathways in toluene photooxidation? Mass fraction can only tell relative change.

**Author reply:**

Fragments derived from the AMS data have been extensively used to explore the bulk compositions and properties of ambient organic aerosols (Ng et al., 2010; Zhang et al., 2011; Liu et al., 2017b). m/z 43 ($C_2H_3O^+$) and m/z 44 ($CO_2^+$) mass concentrations were related to SOA mass concentration. It is obviously more reasonable to compare the SOA formation with the relative content of its chemical composition.

We changed the sentence in line 352-353 "Here, we used f43 and f44 to express the fractions of $C_2H_3O^+$ and $CO_2^+$ to the total organic signal." as "Here, we use the approach of Ng et al. (2010) by plotting the fractions of the total organic signal at m/z 43 *vs*. m/z 44 (f43 *vs*. f44)."

22. In tracking Hi-NOA and LO-NOA via PMF, have you ever considered application of some characteristic fragments indicating products by homogeneous acid-base reactions and heterogeneous Maillard reactions to verify your hypothesis? Molecular analysis by HRMS is most reliable compared to AMS derived results.

**Author reply:**

We are very grateful to the reviewer for this comment.

The heterogeneous reaction of $NH_3$ was obtained based on the evolution of SOA chemical composition in this study. HRMS can only measure the SOA chemical composition offline. Both acid-base reactions and Maillard reactions have been proved by previous studies (Na et al., 2007; De Haan et al., 2009; Huang et al., 2012; Ortiz-Montalvo et al., 2014; Paciga et al., 2014; Chen et al., 2019; Qi et al., 2020). The specific products of toluene/$NH_3$ SOA are not the focus of this study, hence the HRMS was not used here.

m/z 27 ($CHN^+$) and 30 ($CH_4N^+$) are the characteristic fragments of imidazole (Malloy et al., 2009). As shown in Fig. 4, more m/z 27 and 30 were observed in the stable stage.

23. Line 439: why dose Lo-NOA have higher N/C ratio? 0.062 vs. 0.29 for Ho-NOA vs. Lo-NOA?

**Author reply:**

Here is mistake. The N/C ratio of Lo-NOA is 0.029.

It was corrected.

24. Line 449: how did you count the values of N/C ratio and percentage? Check all the values in the script carefully. 0.0404 and 0.216 accounts for 65% and 35 %, respectively?

**Author reply:**

We are very grateful to the reviewer for the attention to the data.

Here is mistake. The contribution of NOx to the N/C value was 0.0216, not 0.216. We have fixed it.

N/C ratio of Hi-NOA and Lo-NOA in the $NH_3$ experiment was 0.032 and 0.010, respectively. N/C ratio of Hi-NOA and Lo-NOA in the NOx experiment was 0.019 and 0.014, respectively. In Exp. 3 with both $NH_3$ and NOx presence, the proportion of NOA formed from $NH_3$ and NOx was defined as σ and 1-σ, respectively. We assumed that

the effects of NH$_3$ or NOx on the N/C ratio in the Hi-NOA and Lo-NOA factors did not change among different experimental conditions. The proportion of NOA formed from NH$_3$ and NOx can be calculated as:

$$\frac{0.010\times\sigma+0.014\times(1\text{-}\sigma)}{0.032\times\sigma+0.019\times(1\text{-}\sigma)} = \frac{0.029}{0.062}$$

In the Lo-NOA factor, the proportion and contribution of N/C from NH$_3$ was calculated as 0.010×σ/[0.010×σ+0.014×(1-σ)] and 0.029×0.010×σ/[0.010×σ+0.014×(1-σ)], respectively.

In the Hi-NOA factor, the proportion and contribution of N/C from NH$_3$ was calculated as 0.032×σ/[0.032×σ+0.019×(1-σ)] and 0.062×0.032×σ/[0.032×σ+0.019×(1-σ)], respectively.

The proportion and contribution of N/C from NOx were calculated in the same way.

25. SOA concentrations have been summarized in Table 1, Figure 1 is not necessary, remove it. Besides, it is suggested to add photon flux and OH radical concentration information in Table 1.

**Author reply:**

We changed Fig.1 as below:

[Figure]

"

[Figure]

**Fig. 1.** The evolution of mass concentration (a) and yield (b) of toluene-derived SOA in different experiments. All the mass concentrations were wall-loss corrected. The error bars of SOA yield were calculated by the volatility of measured SOA concentration after the UV light was turn off at the end of each experiment."

Photon flux was not known here, so it was not shown in the manuscript.

The OH concentration is very important for the photooxidation. We added average OH concentration in line 163: "The average OH concentration over the entire reaction period was 5.87 $\times$ $10^7$ molecule $cm^{-3}$."

26. Redraw Figure 3, make it clearer. Suggest to add time information to trace time-profile of f43-f44

**Author reply:**

We are very grateful to the reviewer for this comment. There are too many points in Fig. 3, and make it confusing. Therefore, we appropriately reduce the number of data points without changing the results. The new Fig. 3 is shown below.

[Figure]

The time-profile of f43 *vs.* f44 was added in the Supporting Information, and shown below.

[Figure]

Fig.S4 Time-profile of f43 *vs.* f44 for different experiments"

27. Make clear statement of specific time for formation and stable stage of SOA in Figure 4, is it stable minus initial stage? Explain the presence of nitrogen-bearing fragments in Experiment 1.

**Author reply:**

The statement for formation and stable stage was added in line 377: "According to the changing trend of SOA concentration over time, the photooxidation process was

divided into formation stage and stable stage. As shown in Fig. 1, the first half hour of photooxidation when SOA concentration increased linearly with time was defined as SOA formation stage. After 60 min of photooxidation, SOA concentration was not changed with reaction time, and it was defined as stable stage."

We also added this sentence in line 815 to explain the presence of nitrogen-bearing fragments in Exp. 1.: "Only minimal N-containing fragments could be observed in the Exp.1 in the absence of either $NH_3$ or NOx. These N-containing fragments could be attributed to the background $NH_3$ and NOx in the chamber or the systematic errors from AMS."

28. Present error and statistic results in Figure 5.

**Author reply:**

There is no error obtained from the PMF result.

We added the statistic results in Fig.5 as below.

[Figure]

**Fig. 5.** The H/C, O/C, and N/C values of Hi-NOA and Lo-NOA for each experiment. (a) Exp. 2 with 200 ppb $NH_3$, (b) Exp. 3 with 200 ppb $NH_3$ and 62 ppb $NO_2$, and (c) Exp. 4 with 63 ppb $NO_2$.

29. Keep axis label and legend consistent in Figure 6.

**Author reply:**

We changed Fig.6 as below. As suggested by Referee #4, the y-axis title was changed as "Percentage in total (%)"

[Figure]

"

**Fig. 6.** The evolution of high-nitrogen OA (Hi-NOA) and low-nitrogen OA (Lo-NOA) during the photooxidation process under different NOx/NH$_3$ concentrations. Hi-NOA and Lo-NOA were not consistent among experiments. (a) Exp. 2 with 200 ppb NH$_3$, (b) Exp. 3 with 200 ppb NH$_3$ and 62 ppb NO$_2$, and (c) Exp. 4 with 63 ppb NO$_2$."

**Referenc:**

Chen, T. Z., Liu, Y. C., Ma, Q. X., Chu, B. W., Zhang, P., Liu, C. G., Liu, J., and He, H.: Significant source of secondary aerosol: formation from gasoline evaporative emissions in the presence of $SO_2$ and $NH_3$, Atmos. Chem. Phys., 19, 8063-8081, 10.5194/acp-19-8063-2019, 2019.

Chen, Y., and Bond, T. C.: Light absorption by organic carbon from wood combustion, Atmos. Chem. Phys., 10, 1773-1787, 10.5194/acp-10-1773-2010, 2010.

Chhabra, P. S., Ng, N. L., Canagaratna, M. R., Corrigan, A. L., Russell, L. M., Worsnop, D. R., Flagan, R. C., and Seinfeld, J. H.: Elemental composition and oxidation of chamber organic aerosol, Atmos. Chem. Phys., 11, 8827-8845, 10.5194/acp-11-8827-2011, 2011.

De Haan, D. O., Corrigan, A. L., Tolbert, M. A., Jimenez, J. L., Wood, S. E., and Turley, J. J.: Secondary organic aerosol formation by self-reactions of methylglyoxal and glyoxal in evaporating droplets, Environ. Sci. Technol., 43, 8184-8190, 10.1021/es902152t, 2009.

Docherty, K. S., Corse, E. W., Jaoui, M., Offenberg, J. H., Kleindienst, T. E., Krug, J. D., Riedel, T. P., and Lewandowski, M.: Trends in the oxidation and relative volatility of chamber-generated secondary organic aerosol, Aerosol Sci. Technol., 52, 992-1004, 10.1080/02786826.2018.1500014, 2018.

Huang, Y., Lee, S. C., Ho, K. F., Ho, S. S. H., Cao, N. Y., Cheng, Y., and Gao, Y.: Effect of ammonia on ozone-initiated formation of indoor secondary products with emissions from cleaning products, Atmos. Environ., 59, 224-231, 10.1016/j.atmosenv.2012.04.059, 2012.

Ji, Y., Zhao, J., Terazono, H., Misawa, K., Levitt, N. P., Li, Y., Lin, Y., Peng, J., Wang, Y., Duan, L., Pan, B., Zhang, F., Feng, X., An, T., Marrero-Ortiz, W., Secrest, J., Zhang, A. L., Shibuya, K., Molina, M. J., and Zhang, R.: Reassessing the atmospheric oxidation mechanism of toluene, Proc. Natl. Acad. Sci. U. S. A., 114, 8169-8174, 10.1073/pnas.1705463114, 2017.

Jiang, X. T., Lv, C., You, B., Liu, Z. Y., Wang, X. F., and Du, L.: Joint impact of atmospheric $SO_2$ and $NH_3$ on the formation of nanoparticles from photo-oxidation of a typical biomass burning compound, Environ Sci-Nano, 7, 2532-2545, 10.1039/d0en00520g, 2020.

Kroll, J. H., Donahue, N. M., Jimenez, J. L., Kessler, S. H., Canagaratna, M. R., Wilson, K. R., Altieri, K. E., Mazzoleni, L. R., Wozniak, A. S., Bluhm, H., Mysak, E. R., Smith, J. D., Kolb, C. E., and Worsnop, D. R.: Carbon oxidation state as a metric for describing the chemistry of atmospheric organic aerosol, Nat. Chem., 3, 133-139, 10.1038/nchem.948, 2011.

Liu, S. J., Jia, L., Xu, Y., Tsona, N. T., Ge, S. S., and Du, L.: Photooxidation of cyclohexene in the presence of $SO_2$: SOA yield and chemical composition, Atmos. Chem. Phys., 17, 13329-13343, 10.5194/acp-17-13329-2017, 2017a.

Liu, S. J., Wang, Y., Wang, G., Zhang, S., Li, D., Du, L., Wu, C., Du, W., and Ge, S.: Enhancing effect of $NO_2$ on the formation of light-absorbing secondary organic aerosols from toluene photooxidation, Sci. Total Environ., 794, 148714, 10.1016/j.scitotenv.2021.148714, 2021.

Liu, T. Y., Wang, X. M., Deng, W., Zhang, Y. L., Chu, B. W., Ding, X., Hu, Q. H., He, H., and Hao, J. M.: Role of ammonia in forming secondary aerosols from gasoline vehicle exhaust, Sci. China Chem., 58, 1377-1384, 10.1007/s11426-015-5414-x, 2015.

Liu, T. Y., Li, Z., Chan, M., and Chan, C. K.: Formation of secondary organic aerosols from gas-phase emissions of heated cooking oils, Atmos. Chem. Phys., 17, 7333-7344, 10.5194/acp-17-7333-2017, 2017b.

Malecha, K. T., and Nizkorodov, S. A.: Photodegradation of secondary organic aerosol particles as a source of small, oxygenated volatile organic compounds, Environ. Sci. Technol., 50, 9990-9997, 10.1021/acs.est.6b02313, 2016.

Malloy, Q. G. J., Li, Q., Warren, B., Cocker Iii, D. R., Erupe, M. E., and Silva, P. J.: Secondary organic aerosol formation from primary aliphatic amines with $NO_3$ radical, Atmos. Chem.

Phys., 9, 2051-2060, 10.5194/acp-9-2051-2009, 2009.

Na, K., Song, C., Switzer, C., and Cocker, D. R.: Effect of ammonia on secondary organic aerosol formation from α-pinene ozonolysis in dry and humid conditions, Environ. Sci. Technol., 41, 6096-6102, 10.1021/es061956y, 2007.

Ng, N. L., Kroll, J. H., Chan, A. W. H., Chhabra, P. S., Flagan, R. C., and Seinfeld, J. H.: Secondary organic aerosol formation from m-xylene, toluene, and benzene, Atmos. Chem. Phys., 7, 3909-3922, 10.5194/acp-7-3909-2007, 2007.

Ng, N. L., Canagaratna, M. R., Zhang, Q., Jimenez, J. L., Tian, J., Ulbrich, I. M., Kroll, J. H., Docherty, K. S., Chhabra, P. S., Bahreini, R., Murphy, S. M., Seinfeld, J. H., Hildebrandt, L., Donahue, N. M., DeCarlo, P. F., Lanz, V. A., Prévôt, A. S. H., Dinar, E., Rudich, Y., and Worsnop, D. R.: Organic aerosol components observed in Northern Hemispheric datasets from Aerosol Mass Spectrometry, Atmos. Chem. Phys., 10, 4625-4641, 10.5194/acp-10-4625-2010, 2010.

Ortiz-Montalvo, D. L., Hakkinen, S. A., Schwier, A. N., Lim, Y. B., McNeill, V. F., and Turpin, B. J.: Ammonium addition (and aerosol pH) has a dramatic impact on the volatility and yield of glyoxal secondary organic aerosol, Environ. Sci. Technol., 48, 255-262, 10.1021/es4035667, 2014.

Paciga, A. L., Riipinen, I., and Pandis, S. N.: Effect of ammonia on the volatility of organic diacids, Environ. Sci. Technol., 48, 13769-13775, 10.1021/es5037805, 2014.

Park, J. H., Babar, Z. B., Baek, S. J., Kim, H. S., and Lim, H. J.: Effects of NOx on the molecular composition of secondary organic aerosol formed by the ozonolysis and photooxidation of α-pinene, Atmos. Environ., 166, 263-275, 10.1016/j.atmosenv.2017.07.022, 2017.

Pathak, R. K., Stanier, C. O., Donahue, N. M., and Pandis, S. N.: Ozonolysis of alpha-pinene at atmospherically relevant concentrations: Temperature dependence of aerosol mass fractions (yields), J Geophys Res-Atmos, 112, Artn D03201, 10.1029/2006jd007436, 2007.

Pieber, S. M., El Haddad, I., Slowik, J. G., Canagaratna, M. R., Jayne, J. T., Platt, S. M., Bozzetti, C., Daellenbach, K. R., Frohlich, R., Vlachou, A., Klein, F., Dommen, J., Miljevic, B., Jimenez, J. L., Worsnop, D. R., Baltensperger, U., and Prevot, A. S. H.: Inorganic salt interference on $CO_2^+$ in Aerodyne AMS and ACSM organic aerosol composition studies, Environmental Science & Technology, 50, 10494-10503, 10.1021/acs.est.6b01035, 2016.

Prinn, R. G., Weiss, R. F., Miller, B. R., Huang, J., Alyea, F. N., Cunnold, D. M., Fraser, P. J., Hartley, D. E., and Simmonds, P. G.: Atmospheric trends and lifetime of $CH_3CCI_3$ and global OH concentrations, Science, 269, 187-192, 10.1126/science.269.5221.187, 1995.

Qi, X., Zhu, S., Zhu, C., Hu, J., Lou, S., Xu, L., Dong, J., and Cheng, P.: Smog chamber study of the effects of NOx and NH3 on the formation of secondary organic aerosols and optical properties from photo-oxidation of toluene, Sci. Total Environ., 727, 138632, 10.1016/j.scitotenv.2020.138632, 2020.

Ruggeri, G., Bernhard, F. A., Henderson, B. H., and Takahama, S.: Model-measurement comparison of functional group abundance in α-pinene and 1,3,5-trimethylbenzene secondary organic aerosol formation, Atmos. Chem. Phys., 16, 8729-8747, 10.5194/acp-16-8729-2016, 2016.

Volkamer, R., Jimenez, J. L., San Martini, F., Dzepina, K., Zhang, Q., Salcedo, D., Molina, L. T., Worsnop, D. R., and Molina, M. J.: Secondary organic aerosol formation from anthropogenic air pollution: Rapid and higher than expected, Geophys. Res. Lett., 33, Artn L17811, 10.1029/2006gl026899, 2006.

Zhang, Q., Jimenez, J. L., Canagaratna, M. R., Ulbrich, I. M., Ng, N. L., Worsnop, D. R., and Sun, Y.: Understanding atmospheric organic aerosols via factor analysis of aerosol mass spectrometry: a review, Anal. Bioanal. Chem., 401, 3045-3067, 10.1007/s00216-011-5355-y, 2011.

Zou, Y., Deng, X. J., Zhu, D., Gong, D. C., Wang, H., Li, F., Tan, H. B., Deng, T., Mai, B. R., Liu, X. T., and Wang, B. G.: Characteristics of 1 year of observational data of VOCs, NOx and O$_3$ at a suburban site in Guangzhou, China, Atmos. Chem. Phys., 15, 6625-6636, 10.5194/acp-15-6625-2015, 2015.

**Response to Reviewers**

Ms. Ref. No.: acp-2021-560

Synergetic effect of NH$_3$ and NOx on the production and optical absorption of secondary organic aerosol formation from toluene photooxidation

Dear Editor:

We greatly appreciate the time and effort that the editor and reviewer spent in reviewing our manuscript. After reading the comments from the reviewers, we have carefully revised our manuscript. All the changes we made are marked in red. Our responses to the comments are itemized below. The referee's comments are in black, authors' responses are in blue.

Anything for our paper, please feel free to contact me via ghwang@geo.ecnu.edu.cn.

All the best

Gehui Wang

Oct. 14, 2021

**Reviewer #2**

This manuscript describes a set of four environmental chamber experiments studying the formation of organic aerosol from the photooxidation of toluene. The authors show that the addition of NOx during photooxidation suppresses SOA formation, while the addition of NH3 enhances SOA formation, consistent with previous studies. They then show that photooxidation of toluene in the presence of both NOx and NH3 together results in an even greater enhancement of SOA formation than NH3 alone. They ascribe this to a "synergistic" effect whereby the more-volatile compounds produced by photooxidation with NOx are able to form SOA efficiently when they can react with NH3 (or NH4+). AMS and optical

observations are also reported and used to support the hypothesis that carbonyl + NH3/NH4+ reactions are responsible for the observed SOA enhancement.

Overall, the study reports an interesting effect that could be important in the atmosphere. The introduction also gives a great summary of all the reactants involved and their expected effects on SOA formation in the atmosphere. However, the methods leave a lot of unanswered questions that make it difficult to know how to interpret results. In particular, the lack of replicate experiments, lack of reported data on the amount of toluene oxidized or its rate of oxidation, and lack of consideration of vapor wall losses and potential variability of particle wall loss, all conspire to introduce a great deal of uncertainty into the results, which are largely reported without any estimates of error or uncertainty. More details on these methodology questions are provided below.

Furthermore, the conditions for the experiments performed here are far removed from the ambient atmosphere and make it difficult to say whether the observed effects would apply in the atmosphere. A number of statements in the paper suggesting that this synergistic chemistry is responsible for ambient observations are therefore not supported by the evidence, e.g. L 255-258, "This may explain why predictions of SOA concentrations in large-scale atmospheric models, which typically describe SOA formation from data derived from chamber experiments, are frequently lower than field observations" and also L 519-520, "In the actual atmosphere, NOx and NH3 co-exist. Therefore, the findings presented here clearly show that the synergetic effects of NOx and NH3 should not be neglected." Alternatively (even better!), more analysis or further experiments should be provided to characterize how the observations herein might be expected to differ when applied to the conditions of the ambient atmosphere, and greater discussion should be included on this point.

I'm generally very hesitant to say more experiments need to be done in the review stage, but unless there are others that have been done already but aren't clearly reported here (or more specific data from the ones that have been performed), I think this is a case where more is needed. There are too many confounding variables here that aren't easily explained away (reproducibility? Dependence on oxidation rate / OH? How much toluene was oxidized in each? Dependence on concentrations? Wall losses? Balance of NO and $NO_2$?). Hopefully the more specific questions below on the methods will provide more detailed guidance on how the manuscript can be improved to help mitigate these uncertainties.

**Author reply:**

We thank the reviewer for the comments above, which are very important and helpful for improving our paper quality and mitigating the related result uncertainties. We have carefully revised our work. Followings are the detailed answers.

**General questions on methods:**

**1.** Were any replicate experiments performed at the separate reaction conditions, and if so, how reproducible were the results? The only uncertainty estimates I see are on temp/RH (how are those arrived at) and SOA formed which I assume is from SMPS measurements (how is uncertainty calculated from that?), but doesn't tell us anything about reproducibility of the results from experimental variability, e.g. of the reactant injections and wall losses. Without this, it's difficult to know what uncertainty to ascribe to the results.

**Author reply:**

We express our appreciation to the reviewers for the comments.

The repeated experiments were not carried out here. The error bars in Fig. 1 were calculated by the fluctuation of SOA concentration when the UV light was turn off at the end of each photooxidation experiment. In order to make this section clearer, the sentences of "The error bars were calculated by the fluctuation of measured SOA concentration after the UV light was turn off at the end of each experiment." was added in the caption of Fig.1.

**2.** Do you have any way of knowing how much toluene was oxidized in each experiment, and how quickly (i.e. how much OH there was)? This would be nice for converting from the SOA mass formed that you report to an SOA mass yield, which would be easier to compare to previous studies. It would also help to know whether differences in OH (and therefore the rate of oxidation and the peak RO2 radical production) could explain some difference in the observed SOA formed and the SOA properties, as has been shown in the past.

**Author reply:**

We are very grateful to the reviewer for this comment. We measured the toluene concentration with PTR-MS on-line. The evolution of toluene concentration at different experiment conditions was shown below and add in the Supporting Information. The evolution of toluene concentrations was almost uniformly at different NOx and NH$_3$ conditions.

[Figure]

Fig.S1 The evolution of toluene concentration for each experiment.

In the revised manuscript, we have added the sentences in line 163 as follows: "Toluene concentration was measured with a Proton Transfer Reaction-Mass Spectrometry (PTR-ToF-MS, Ionicon Analytik, Austria). The evolution of toluene concentration for different experiments was shown in Fig.S1."

we also added the sentences of "SOA yield (Y) is defined as $Y = \Delta M_0/\Delta HC$, where $\Delta M_0$ is the produced organic aerosol mass concentration (µg m$^{-3}$), and $\Delta HC$ is the mass concentration of reacted toluene (µg m$^{-3}$)." in line 235.

"The SOA mass concentrations at different conditions are shown in Fig. 1." in line 234 was changed as "The evolution of SOA mass concentrations and SOA yield at different conditions during the photooxidation process were shown in Fig.1"

Table 1 was also updated as below:

**Table 1.** Summary of experimental conditions in this study

| No. | $Tol_0$ (ppb) | $\Delta$ Tol (ppb) | $NH_3$[a] (ppb) | $NO_2$ (ppb) | RH (%) | SOA mass conc.[b,c] ($\mu$g m$^{-3}$) | SOA yield[b] (%) |
|---|---|---|---|---|---|---|---|
| Exp.1 | 664.1 | 551.2 | - | - | 25 ± 1 | 637 ± 14.6 | 28.1 |
| Exp.2 | 618.7 | 499.4 | ~200 | - | 23 ± 1 | 867 ± 12.7 | 34.7 |
| Exp.3 | 620.9 | 516.1 | ~200 | 62 | 26 ± 1 | 1020 ± 10.6 | 42.7 |
| Exp.4 | 645.7 | 532.5 | - | 63 | 25 ± 1 | 452 ± 18.9 | 19.5 |

[a] The concentration of $NH_3$ is estimated by the amount of $NH_3$ added and the volume of the smog chamber. [b] SOA concentration and yield were calculated after taking into account the wall loss. [c] The reported SOA mass concentrations is the peak values after the wall loss correction.

In the revised manuscript, we have added the description of OH concentration calculation in line 163 as follows: "OH concentration in the chamber was calculated based on the first order decay of toluene concentration. There was no obvious difference of OH concentrations in the different NOx and $NH_3$ levels (Fig. S2)."

The highest OH concentration of $1.02 \times 10^8$ molecule cm$^{-3}$ was observed at the beginning of the reaction. The average OH concentration over the entire reaction period was $5.87 \times 10^7$ molecule cm$^{-3}$. The calculation of OH concentration was added in the Supporting Information as below:

**"S1 OH Concentration Calculation Process**

The OH concentration was calculated based on the decay ratio of toluene concentrations and the known rate constant with respect to OH. The change of toluene concentration over time can be expressed as:

$$-\frac{d[\text{toluene}]}{dt} = K_{\text{OH}} \times [\text{OH}] \times [\text{toluene}] \quad \text{(RS1)}$$

Where, $K_{OH}$ is the reaction rates constant of OH radicals with toluene ($K_{OH}$=5.7×10$^{-12}$ cm$^3$ molecule$^{-1}$ s$^{-1}$). Assuming that the concentration of hydroxide did not change during the experiment, then we can get:

$$\ln\left(\frac{[toluene]_0}{[toluene]_t}\right)/t = K_{OH} \times [OH] \qquad \text{(RS2)}$$

Thus, plotting the variation curve of $\ln([toluene]_0/[toluene]_t)$ vs. time t showed as Fig.S1. The $\ln([toluene]_0/[toluene]_t)$ in Fig.1(b) was not a straight line. This is because the OH is consumed as the reaction goes on. The evolution of OH concentration at experiment conditions was shown in Fig.S2. The different experiment conditions in this study did not affect the OH concentration obviously. The highest OH concentration of $1.0 \times 10^8$ molecule cm$^{-3}$ was observed at the beginning of the reaction. The average OH concentration over the entire reaction period is $5.9 \times 10^7$ molecule cm$^{-3}$.

[Figure]

Fig.S2 The evolution of OH concentrations at different experiment conditions.

**3.** Do you know, either from the AMS or a control experiment without toluene, how much inorganic aerosol was formed from the interaction of ammonium and NOx in the chamber? It seems this could explain a large portion of the "enhancement" in particle mass in the NH3 + NOx experiment. It could also affect gas-particle partitioning by acting as a "seed" aerosol on which more vapors can condense (which is a form of NOx-NH3 synergy, I suppose, but seems less chemistry-dependent than the effects to which you ascribe the synergy).

**Author reply:**

The concentration variations of $NO_3^-$ and $NH_4^+$ during the experiments were measured by the AMS. The content of $NO_3^-$ and $NH_4^+$ is very low in particulate matter. The content of $NO_3^-$ and $NH_4^+$ for each experiment was list in the table below.

**Table S1.** The content of $NO_3^-$ and $NH_4^+$ in the particle-phase for each experiment

|  | $[NOx]_0$ (ppb) | $[NH_3]_0$ (ppb) | $[NO_3^-]/[Org]$ (%) | $[NH_4^+]/[Org]$ (%) |
|---|---|---|---|---|
| Exp.2 | - | ~200 | - | 1.9 |
| Exp.3 | 62 | ~200 | 4.0 | 2.6 |
| Exp.4 | 63 | - | ＜0.2 | - |

As seen in Table S1, when only $NO_2$ was introduced into the chamber, the content of $NO_3^-$ in particulate matter was less than 0.2%. Based on the consumption of $NO_2$, most of the $NO_3^-$ was present in the gas phase in the form of nitric acid. When only $NH_3$ was introduced in the chamber, the content of $NH_4^+$ was about 1.9%. When both $NO_2$ and $NH_3$ were simultaneously introduced into the chamber, the $NO_3^-$ reacted with $NH_4^+$ to form ammonium nitrate particles, which accounted for 6.6% of the total mass of particulate matter in the chamber. AMS cannot distinguish whether $NO_3^-$ and $NH_4^+$ come from organic or inorganic phase, but 6.6% is obviously the upper limit of inorganic components in particulate matter if we assume that both are entirely derived from inorganic phase. The particle mass increased about 59% in the $NH_3$ + NOx experiment compared that with no $NH_3$ or NOx, which cannot be explained by the formation of inorganic $NH_4NO_3$ but can only be ascribed to the synergetic effect of $NH_3$ and NOx on the toluene SOA formation. The following sentences were added in line 253 of the manuscript to explain the possible influence of the smaller amount of the inorganic aerosol on the enhancement in particle mass in the $NH_3$ + NOx experiment.

"Although inorganic aerosol was formed from the interaction of $NH_3$ and NOx in the chamber, the upper limit of the inorganic matter only account for 6.6% of the total mass of particulate matter (Table S1) in the $NH_3$ + NOx experiment. Therefore, it was not the main cause of the increase in particulate matter."

**4.** - Were gas-phase reactant concentrations measured during experiments? In addition to helping determine how much toluene was oxidized and how quickly during each experiment (see above), this would be really helpful for knowing the balance of NO2 and NO in the high-NOx experiments. These two NOx compounds have very different effects on gas-phase chemistry, and even if they do interconvert with the lights on, it's important to understand their relative abundances to know how much RO2 reacts with NO, how much HNO3 is formed, and whether PANs form from the reactions of acyl peroxy radicals with NO2, for example.

**Author reply:**

We express our gratitude to the reviewer for this comment. The gas-phase reactant concentrations were measured (see our reply to the reviewer comments #2 above). NOx concentration is one order of magnitude lower than toluene concentration, and this may be the reason why NOx has no obvious effect on OH concentration.

In the atmosphere, nitrogen oxides are emitted from natural and anthropogenic sources primarily as NO, which rapidly achieves steady state with $NO_2$. In the presence of VOCs, $NO_2$ can be lost by reaction with $RO_2$ to form peroxy nitrates ($RO_2NO_2$) (O'Brien et al., 1995). $RO_2NO_2$ species are thermally unstable at boundary layer temperatures and decompose back to $NO_2$ and $RO_2$ on a timescale of minutes (Fisher et al., 2016). $RONO_2$ species formed form $RO_2 + NO$ can dominate NOx loss (Zhao et al., 2018;Browne et al., 2014). Longer-lived peroxyacylnitrates (PANs) can also formed through $RO_2 + NO_2$ reaction, but Mao et al. (2013) pointed PANs were less efficient for reactive N export than $RONO_2$. All in all, although $NO_2$ is a major component of NOx in the chamber, the reaction of $RO_2$ with NO is still the main process for $RO_2$ consumption.

**5.** - Why were such high concentrations of toluene used, and how much could RO2 + RO2 chemistry participate, especially in the low-NOx experiment? How does this compare to the amount of toluene-RO2 that are expected to react with toluene-RO2 in ambient conditions in the atmosphere? It seems that excessive RO2-RO2 chemistry could make these "low-NOx" results unrepresentative of "low-NOx" conditions in the atmosphere, making interpretation difficult.

**Author reply:**

We fully agree with the reviewer on this opinion. Considering the performance of this chamber, as well as to obtain more accurate analyzation of chemical composition, a higher concentration of toluene was used in this study. The atmosphere is very complicated, and the experimental conditions in the chamber cannot be the same as the real environment. In our work, the proportion of the key components is comparable to the ambient level. The mean concentration of aromatics VOCs and OH was about 11 ppb (Zou et al., 2015) and $1 \times 10^6$ molecule $cm^{-3}$ (Prinn et al., 1995), respectively, which was consist with the VOCs/OH ratio (~700 ppb/$5.9 \times 10^7$ molecule $cm^{-3}$) in our study.

We thank the Referee for this insightful comment. We also think the quantitative of $RO_2 + RO_2$ chemistry is very important. Unfortunately, we do not know how much could $RO_2 + RO_2$ chemistry participate at different NOx conditions. As suggested by the reviewer, $RO_2 + RO_2$ chemistry is excessive when NOx was not added into the chamber. But the original objective of our study was not to determine the property of $RO_2 + RO_2$ and $RO_2 + NO$ on toluene SOA formation. The different "low-NOx" conditions between chamber and real atmosphere does not affect the purpose of chamber experiment to obtain the effect of NOx on the photooxidation mechanism.

To provide a better illustration of the experiment condition, the following sentences have been added in the revised manuscript in line 170:

"In our work, the OH and toluene concentrations were higher than those of urban conditions. The purpose of the high OH and toluene concentrations is to obtain enough particle production samples for off-line collections and accurate measurements. The toluene concentrations remained stable under the different experimental conditions, the variation of toluene-derived SOA mass concentration and yield was only affected by the different $NO_2$ and/or $NH_3$ concentrations in this study. Toluene was studied here as the representative of total aromatic VOCs in the urban atmosphere. The concentration

ratio of toluene to OH in this study is similar to that under the real atmospheric conditions (Zou et al., 2015;Prinn et al., 1995)."

The follow sentences were also added in line 523 of the revised manuscript: "It has to be noted that the concentration of reactants used for the experiments is much higher than that observed in polluted areas, the effect of $NH_3$ and NOx on the photooxidation of toluene with lower concentrations would be checked in the further study."

**6.** - Particle wall losses are known to be affected by both particle size and composition, both of which evolve between and over the course of your experiments (as might the surface-area-to-volume ratio of the chamber). It seems the use of a single particle wall loss rate across all sizes and across experiments could introduce substantial error or at least uncertainty to the calculations of particle mass formed. Even if it proves too difficult to quantify size- and composition-dependent wall loss rates, more discussion is needed regarding the limitation of this method.

**Author reply:**

The presence of $NH_3$ could lead to an increase in particle wall loss ratio. All the particle mass concentration was corrected with the same way of Jiang et al. (2020) and Pathak et al. (2007) to constrained the influence of wall losses of different SOA formed with different experiment conditions. For each experiment, we continued to monitor the particle concentration in the dark condition for 1 hour, and recalculated the particle wall loss constant according to the variation of particle concentration.

The following was added at line 168 for clarification.

"However, the particle wall loss rates were detected at the end of the chamber experiment after the UV-lamps were turned off, and the mass concentration was corrected with the same way of Jiang et al. (2020) and Pathak et al. (2007)"

**7.** - Similarly, vapor wall losses are likely to play a major role in these experiments, especially in a relatively small chamber, and it's been shown they

can bias measured SOA yields from toluene when no seed aerosol is used (see https://acp.copernicus.org/articles/15/4197/2015/ and https://www.pnas.org/content/111/16/5802). They will also vary over the course of the experiment, and could differ between experiments if (a) the reactive intermediates formed have different volatilities, and (b) the particles are formed at different rates and therefore the semivolatile vapors experience different wall-vs-particle competition for partitioning. How might this affect the results? Seeded experiments would be particularly useful here to understand these effects.

**Author reply:**

We agree with the Referee that the wall loss rates are also different dependent on vapor wall losses. The walls serve as a large reservoir of equivalent OA mass that compete with the particulate SOA. But the wall loss effect of gas-phase products on SOA formation have not yet been quantitatively established. After the wall loss correction, the particle mass concentration was almost constant (New Fig.1), we believe that our results are reliable and credible.

To clarify the statement, we modified the sentences in the revised manuscript. The new one reads as follows:

"Recent experiments shown that the wall loss of organic vapors to the Teflon walls should not be ignored (Zhang et al., 2014;Zhang et al., 2015), and represented a major challenge in investigating SOA formation with environmental chambers (Zhang et al., 2014;Krechmer et al., 2020). The formation of SOA in laboratory chambers may be substantially suppressed due to losses of SOA-forming vapors to chamber walls, but this effects on SOA formation have not yet been quantitatively established. However, the particle wall loss rates were detected at the end of the chamber experiment after the UV-lamps were turned off, and the mass concentration was corrected with the same way of Jiang et al. (2020) and Pathak et al. (2007). After the wall loss correction, the particle mass concentration was almost constant, the different wall loss effect caused by gaseous oxidation products formed in the different experiment conditions was considered remedied."

We also discussed the wall loss in the section 3.1 and added the following sentence at line 235.

"Interestingly, wall loss is increased 66% and 205% in Exp.2 (in the presence of NH3) and Exp.3 (in the mixed condition of $NH_3$ and NOx), respectively, when compared with the experiments with no $NH_3$ (Exp.1 and 4). The larger particle wall loss in the presence of $NH_3$ could be explained by increasing condensation process of oxidized organic vapors onto the Teflon chamber wall via oligomerization (for dicarbonyls) and ionic dissociation/acid-base reaction (for organic acids)."

As pointed by the Referee the seed particles in the chamber provided adequate seed surface area at the beginning of the reaction, which constrained the influence of vapor wall losses of the semi-volatile vapor. Because of the high concentrations of toluene and $H_2O_2$ in this study, the maximum number concentration and maximum particle surface area is $4.6\text{-}5.7 \times 10^5$ $cm^{-3}$ and $1.3\text{-}1.8 \times 10^4$ $\mu m^2$ $cm^{-3}$, respectively. The resulting total AS seed surface area in the study of the Zhang et al. (2014) ranged from $\sim 1 \times 10^3$ $\mu m^2$ $cm^{-3}$ up to $\sim 1 \times 10^4$ $\mu m^2$ $cm^{-3}$. The loss of condensable vapors to the chamber walls leads to a low bias in the observed SOA formed even for the experiments with the highest seed SA. In their study, the vapor wall loss bias ($R_{wall}$) did not change much when the seed surface area is greater than $5.5 \times 10^3$ $\mu m^2$ $cm^{-3}$. Meanwhile, the calculated $R_{wall}$ varies with oxidant and precursor concentration (actually, VOC loss rate), smaller $R_{wall}$ is obtained when oxidation is faster and precursor VOC concentration is larger. We believe that the addition of seed particles would not impact on SOA formation significantly in this study.

We also need to emphasize that, in future studies, especially when the SOA formation is studied with low VOCs and OH concentrations, seeded experiments would be very useful for us to understand vapor wall losses at different experiment conditions, and improve the accuracy of SOA formation in the chamber study.

**8.** - Is the chamber operated in batch mode, or with a continuous dilution flow during experiments? If there's no flow, how might the changing bag volume and shape affect particle or vapor wall losses? If there is a dilution flow, how was this accounted for in calculations of particle mass concentration, and in terms of the dilution affecting gas-particle partitioning of semivolatile vapors?

**Author reply:**

This chamber was operated in batch mode, and no dilution flow was used here, because the Teflon bag is flexible and thus the volume is variable. The total sampled volume for the online measurement was only 3 L min$^{-1}$. The illumination time of UV light was 120 min. The total sample volume was lower than 8% of the total volume. The change rate of chamber volume is even lower than the systematic error of SMPS (10%) (Liu et al., 2017). Therefore, the change of chamber volume does not significantly affect particle or vapor wall loss during the whole photooxidation process of each experiment. The volume and the change rate of chamber volume were consistent in each experiment. Therefore, particle or vapor wall losses affect by the changing volume of the chamber will not introduce much uncertainties in one experiment or between all the experiments in the present study.

**9.** - 254 nm lights are fairly high-energy / low-wavelength -- does this mean the experiments had extremely high OH? Does it potentially cause rapid photolysis of highly photolabile compounds (like hydrooperoxides and dicarbonyls), and how could that cause differences between chamber conditions and ambient environmental conditions?

**Author reply:**

The photolysis of gas-phase $H_2O_2$ through the 254 nm was wildly used for the OH radical production (Ng et al., 2007;Liu et al., 2021;Jiang et al., 2020). As mentioned before, the average OH concentration in the chamber over the entire reaction period is $5.87 \times 10^7$ molecule cm$^{-3}$. The high OH concentration was due to the high concentration of $H_2O_2$ (about 2 ppm) in the chamber. The UV-light intensity was not very strong. The direct photolysis can be neglected relative to the OH oxidation.

In the troposphere, main oxidation reaction of toluene was initiated by OH (Atkinson and Arey, 2003). The OH oxidation in this study was consistent with the atmospheric oxidation process of toluene in the real atmosphere.

**10.** - Finally, are the reported SOA mass concentrations peak values or at the end of each experiment? It would be good to see their evolution vs. time, to understand how that differs between experiments and whether that could influence results.

**Author reply:**

The reported SOA mass concentrations was the peak values after the wall loss correction. Meanwhile, as shown in the Fig.1, SOA mass concentration is almost unchanged until the end of the reaction. The maximum concentration of SOA is the same as the concentration at the end of the reaction. We also added the sentences after Table 1: "The reported SOA mass concentrations was the peak values after the wall loss correction."

The sentence of "The SOA mass concentrations at different conditions are shown in Fig. 1." in line 234 was changed as "SOA yield (Y) is defined as $Y = \Delta M_0 / \Delta HC$, where $\Delta M_0$ is the produced organic aerosol mass concentration ($\mu g\ m^{-3}$), and $\Delta HC$ is the mass concentration of reacted toluene ($\mu g\ m^{-3}$). The evolution of SOA mass concentrations and SOA yield at different conditions during the photooxidation process were shown in Fig.1."

The SOA mass concentration evolution vs. time was added in Fig.1.

[Figure]

[Figure]

Fig. 1. The evolution of mass concentration (a) and yield (b) of toluene-derived SOA in different experiments. All the mass concentrations were wall-loss corrected. The error bars of SOA yield were calculated by the volatility of measured SOA concentration after the UV light was turn off at the end of each experiment.

**Additional (mostly minor) comments:**

L 65: comma should be a period & start a new sentence

**Author reply:**

Corrected.

L 101-103: Needs more detail. Is this NOx dependence for toluene SOA specifically, or from other precursors? And does "initially" mean early (temporally) in an experiment, or does it refer to slight increases in NOx concentration? Finally, is it all due to the RO2 chemistry as the following sentence suggests, or can some of the increase be ascribed to changing OH as NOx increases?

**Author reply:**

For clarification, the sentence "Laboratory experiments have found that SOA formation was initially enhanced, but then suppressed with increasing NOx concentrations (Sarrafzadeh et al., 2016; Yang et al., 2020)." in line 101-103 is changed to "A clear increase at first and then a decrease in the SOA yield was found with increasing NOx concentration from the laboratory experiments with both artificial (trimethylbenzene) and biological (β-pinene) VOCs (Sarrafzadeh et al., 2016;Yang et al., 2020)."

we also changed the sentences in line 111-113 as follows: "In addition, the increase of OH concentration formed through NO + HO$_2$ →NO$_2$ + OH reaction at low-NOx conditions, and a suppressing effect of NOx on OH formation under high-NOx conditions was partly responsible for the first increasing and then decreasing trend of SOA yield with NOx concentration (Bates et al., 2021;Sarrafzadeh et al., 2016)."

L 107: "participated" should be "precipitate"

**Author reply:**

Suggestion taken.

L 111-112: The change in OH with NOx is not as simple as "suppression", and in many cases NOx may increase OH (see, e.g., Figure 8 in https://acp.copernicus.org/preprints/acp-2021-605/).

**Author reply:**

We thank the Referee for this insightful comment. As mentioned before, we have changed the sentences in line 111-113 as follows: "In addition, the increase of OH concentration formed through NO + HO$_2$ →NO$_2$ + OH reaction at low-NOx conditions, and a suppressing effect of NOx on OH formation under high-NOx conditions was partly responsible for the first increasing and then decreasing trend of SOA yield with NOx concentration (Bates et al., 2021;Sarrafzadeh et al., 2016)"

L 123: unclear; does this mean a reduction in NH3 improves PM2.5 pollution *more than* a reduction in SO2? Is that when reducing both by the same amount?

**Author reply:**

For clarification, the sentence "Indeed, one study observed that a reduction of NH$_3$ emissions improved PM$_{2.5}$ pollution compared to SO$_2$ in winter (Erisman and Schaap, 2004)." in line 101-103 is changed to "Indeed, field observation and model simulation

have pointed out that the reduction of NH₃ emissions contribute much to the improvement of $PM_{2.5}$ pollution compared to $SO_2$ in winter (Erisman and Schaap, 2004)."

L 238: Are these previous studies on toluene or other precursors? How does the magnitude of your SOA-enhancement due to NH3 addition compare to these previous studies?

**Author reply:**

In the study of Qi et al., toluene was used as the SOA precursor. But α-pinene was used in the study of Na et al. We wanted to describe the promotion of NH₃ was observed in both anthropogenic and biological SOA.

In the study of Qi et al., the presence of NH₃ resulted in a 72% increase in SOA mass concentration. However, 200 ppb of NH₃ was injected into the chamber approximately 1.5 h after the UV lights were turned on when the SOA formation reached the maximum value. Because of the different experimental conditions and methods, we don't think the quantitative comparison of previous study to ours is necessary here.

For clarification, we delate the reference of Na et al. (2007) and add the reference of Chu et al. (2016). The sentence of "consistent with previous studies (Na et al., 2007; Qi et al., 2020)." in line 238 was deleted, and "which was consistent with previous studies (Qi et al., 2020;Chu et al., 2016)." was added after the sentence of "There was a noticeable increase in the SOA mass concentration in the presence of NH₃." in line 236

L 244-247: Same comment -- are these previous studies on toluene as well? How does the magnitude of your observed SOA-suppression due to NOx addition compare to the magnitudes observed in these previous studies?

**Author reply:**

BVOC was used in the studies of Zhao et al. (α-pinene and limonene) and Liu et al. (cyclohexene), benzene was used in the study of Xu et al.

The branching of $RO_2$ loss among different pathways has an important influence on the product distribution and thus on SOA composition, physicochemical properties, and yields. The reaction of $RO_2 + NO$ to form the RO intermediate is ubiquitous and not affected by the type of SOA precursors. The fate of $RO_2$ mainly depends on the concentrations of NOx. High NOx can make the $RO_2$ radical fate dominated by one single pathway (i.e., $RO_2 + NO$ or $RO_2 + NO_2$). As mentioned before, the different experimental conditions and methods were used between previous studies and ours. Therefore, we did not compare the quantitative impact of NOx on SOA in different studies.

The following sentences were added in line 253 of the manuscript "The branching of $RO_2$ loss among different pathways has an important influence on the products distribution and SOA formation. The fate of $RO_2$ mainly depends on the concentrations of NOx (Zhao et al., 2018;Liu et al., 2019;Xu et al., 2020)."

L 248: "participate" should be "precipitate"

**Author reply:**

Suggestion taken.

L 258-260: It's not entirely accurate to say this combination hasn't been done before, because SOA-formation experiments have been performed with and without NOx and with and without seed aerosol that usually contains ammonium. If the "synergistic" mechanism you propose (carbonyls+NH4(+) in the aerosol phase) is occurring, that may have been seen before in studies investigating toluene SOA formation with ammonium sulfate seed aerosol. Are there any such studies you can reference and compare to?

**Author reply:**

Ammonium sulfate particle is acidic. It is well known that acid-catalyzed reactions have an important influence on SOA formation (Jang et al., 2002). Joint impact of NOx and NH$_3$ on SOA formation has been studied before (Li et al., 2018;Qi et al., 2020). The following sentences were added in line 255 of the manuscript "Qi et al. (2020) observed the promotion of NH$_3$ on toluene SOA formation was more obviously under high NOx concentration, SOA yield increased 3.7% and 4.6% for 70 ppb and 160 ppb initial NOx concentration, respectively, when 200 ppb NH$_3$ was added into the chamber. Li et al. (2018) shown that the presence NH$_3$ can promote the particle size growth of SOA; at the same time, this particle growth rate was higher under low VOC/NOx (or high NOx) conditions. All in all, the joint effect of multiple environmental factors on SOA formation is not the simple summation of the influences of various factors on SOA formation."

Here, we used "not well-characterized" in the manuscript. For clarify, the sentence of "The effects of multiple factors are not well-characterized by chamber experiments, which was partly responsible for the gap between the simulations and field observations." was delated.

L 295: "even there" -- missing a word, maybe? Does this mean "even if there" or "even when there" (i.e. is it hypothetical, or do you know at some point the OH in the chamber was zero?

**Author reply:**

The OH was zero is hypothetical. The real OH concentration was shown in comment #2. The expression here is not accurate. We express our gratitude for this suggestion.

"even there" was fixed as "even if there".

L 297-300: Unclear if these sentences refer to observations from the current study, or to hypotheticals, or to the cited study by Malecha and Nizkorodov -- was mass loss observed in these experiments, and do you know that OVOCs were photoproduced, and that they had a lower OSc?

**Author reply:**

These sentences refer to the cited study by Malecha and Nizkorodov. In their study, the calculation of SOA particles loss was about ~1% based on the measured OVOC emission rates during summertime conditions in Los Angeles, California. The loss of SOA was very small.

The sentences in line 295-300 of the manuscript were fixed as: "Finally, as pointed by Malecha and Nizkorodov (2016), even if there was no OH in the chamber, the photodegradation of SOA can produce small oxygenated volatile organic compounds (e.g. acetaldehyde $OS_C$=-1, and acetone $OS_C$≈-1.3) under UV light irradiation. The photoproduction of OVOCs from SOA had a lower $OS_C$ value than that of SOA. Although the loss of SOA through photodegradation is small, the $OS_C$ value of SOA still had increased to a certain extent (Malecha and Nizkorodov, 2016)."

L 318-320: Which experiment does this sentence refer to?

**Author reply:**

For clarification, the sentence in line 318-320 is changed to "After 60 min of UV light irradiation, there was no more SOA formation; however, the $OS_C$ did decrease slightly in Exp.2 and 3, illustrating that the $NH_3$ could continue to react with SOA through heterogeneous processes."

L 338: both instances of "volatile" should be "volatility"

**Author reply:**

Corrected.

**Author reply:**

Yes, we corrected.

.

L 398: why tentative? More detail is needed here (as opposed to the SI) about how the PMF analysis was performed (was it performed on the whole dataset from all experiments, or separately on each experiment?) and what the PMF analysis identified -- it seems from the later discussion that it identified two OA factors with differing N/C values, but that should be clearly stated here.

**Author reply:**

We change the sentences in line of 396-400 as below: "For the toluene OH-photooxidation experiments with $NH_3$ and/or NOx presence, two factors were identified from the PMF analysis in the same way of Chen et al. (2019). The H/C, O/C, and N/C values of these two factors are shown in Fig. 5. The factor with higher N/C values was defend as high-nitrogen OA (Hi-NOA). In contrast, the factor with lower N/C values was defend as low-nitrogen OA (Lo-NOA)."

L 414-416: Why is this likely? Because it did not start forming right away? Couldn't that just be because it was a later-generation gas-phase product that reacted with NH3 in the gas phase?

**Author reply:**

As we discussed below, if the Hi-NOA formed from the later-generation gas-phase product, the time-dependent concentrations of Lo-NOA (earlier generational products) should continue to increase. But they exhibited a decline of Lo-NOA at longer reaction times, reflecting the conversion from Lo-NOA to Hi-NOA.

We deleted the sentence of "It was likely that the formation pathway of Hi-NOA did not involve the reaction of $NH_3$ with organic matter in the homogeneous gas phase."

This part has been revised on line 416-422.

"The Lo-NOA reached the maximum mass concentration after 30 min of the photooxidation, and then decreased. Such a decline trend of Lo-NOA at longer reaction times reflected the conversion of Lo-NOA into something else in the particle-phase. As the Lo-NOA decreased, the mass concentration of Hi-NOA gradually increased. Thus, the Hi-NOA should be derived from the heterogeneous reaction of Lo-NOA with $NH_3/NH_4^+$. At the same time, it was proved that the formation pathway of Hi-NOA was not through reaction of $NH_3$ with later-generation gas-phase products in the homogeneous gas phase."

.

L 509: "high volatile" should be either "highly volatile" or "high volatility"

**Author reply:**

Suggestion taken. "high volatile" was fixed as "highly volatile".

L 511: "participation" should be "precipitation"

**Author reply:**

Suggestion taken .

L 526: Data should be put somewhere publicly accessible like an online repository, or, per the ACP guidelines, "If the data are not publicly accessible, a detailed explanation of why this is the case is required." Particularly for these unique experiments I imagine researchers may want to be able to access the data

**Author reply:**

We express our appreciation for this suggestion. Our date will put online soon.

**References**

Atkinson, R., and Arey, J.: Atmospheric degradation of volatile organic compounds, Chem. Rev., 103, 4605-4638, 10.1021/cr0206420, 2003.

Bates, K., Jacob, D., Li, K., Ivatt, P., Evans, M., Yan, Y., and Lin, J.: Development and evaluation of a new compact mechanism for aromatic oxidation in atmospheric models, Atmos. Chem. Phys. Discuss., 2021, 1-34, 10.5194/acp-2021-605, 2021.

Browne, E. C., Wooldridge, P. J., Min, K. E., and Cohen, R. C.: On the role of monoterpene chemistry in the remote continental boundary layer, Atmos. Chem. Phys., 14, 1225-1238, 10.5194/acp-14-1225-2014, 2014.

Chen, T. Z., Liu, Y. C., Ma, Q. X., Chu, B. W., Zhang, P., Liu, C. G., Liu, J., and He, H.: Significant source of secondary aerosol: formation from gasoline evaporative emissions in the presence of $SO_2$ and $NH_3$, Atmos. Chem. Phys., 19, 8063-8081, 10.5194/acp-19-8063-2019, 2019.

Chu, B. W., Zhang, X., Liu, Y. C., He, H., Sun, Y., Jiang, J. K., Li, J. H., and Hao, J. M.: Synergetic formation of secondary inorganic and organic aerosol: effect of $SO_2$ and $NH_3$ on particle formation and growth, Atmos. Chem. Phys., 16, 14219-14230, 10.5194/acp-16-14219-2016, 2016.

Erisman, J. W., and Schaap, M.: The need for ammonia abatement with respect to secondary PM reductions in Europe, Environ. Pollut., 129, 159-163, 10.1016/j.envpol.2003.08.042, 2004.

Fisher, J. A., Jacob, D. J., Travis, K. R., Kim, P. S., Marais, E. A., Chan Miller, C., Yu, K., Zhu, L., Yantosca, R. M., Sulprizio, M. P., Mao, J., Wennberg, P. O., Crounse, J. D., Teng, A. P., Nguyen, T. B., St. Clair, J. M., Cohen, R. C., Romer, P., Nault, B. A., Wooldridge, P. J., Jimenez, J. L., Campuzano-Jost, P., Day, D. A., Hu, W., Shepson, P. B., Xiong, F., Blake, D. R., Goldstein, A. H., Misztal, P. K., Hanisco, T. F., Wolfe, G. M., Ryerson, T. B., Wisthaler, A., and Mikoviny, T.: Organic nitrate chemistry and its implications for nitrogen budgets in an isoprene- and monoterpene-rich atmosphere: constraints from aircraft (SEAC4RS) and ground-based (SOAS) observations in the Southeast US, Atmos. Chem. Phys., 16, 5969-5991, 10.5194/acp-16-5969-2016, 2016.

Jang, M., Czoschke, N. M., Lee, S., and Kamens, R. M.: Heterogeneous atmospheric aerosol production by acid-catalyzed particle-phase reactions, Science, 298, 814-817, 10.1126/science.1075798, 2002.

Jiang, X. T., Lv, C., You, B., Liu, Z. Y., Wang, X. F., and Du, L.: Joint impact of atmospheric $SO_2$ and $NH_3$ on the formation of nanoparticles from photo-oxidation of a typical biomass burning compound, Environ Sci-Nano, 7, 2532-2545, 10.1039/d0en00520g, 2020.

Krechmer, J. E., Day, D. A., and Jimenez, J. L.: Always lost but never forgotten: Gas-phase wall losses are important in all teflon environmental chambers, Environ. Sci. Technol., 54, 12890-12897, 10.1021/acs.est.0c03381, 2020.

Li, K., Chen, L., White, S. J., Yu, H., Wu, X., Gao, X., Azzi, M., and Cen, K.: Smog chamber study of the role of $NH_3$ in new particle formation from photo-oxidation of aromatic hydrocarbons, Sci. Total Environ., 619-620, 927-937, 10.1016/j.scitotenv.2017.11.180, 2018.

Liu, S. J., Jia, L., Xu, Y., Tsona, N. T., Ge, S. S., and Du, L.: Photooxidation of cyclohexene in the presence of $SO_2$: SOA yield and chemical composition, Atmos. Chem. Phys., 17, 13329-13343, 10.5194/acp-17-13329-2017, 2017.

Liu, S. J., Jiang, X. T., Tsona, N. T., Lv, C., and Du, L.: Effects of NOx, $SO_2$ and RH on the SOA formation from cyclohexene photooxidation, Chemosphere, 216, 794-804, 10.1016/j.chemosphere.2018.10.180, 2019.

Liu, S. J., Wang, Y., Wang, G., Zhang, S., Li, D., Du, L., Wu, C., Du, W., and Ge, S.: Enhancing effect of $NO_2$ on the formation of light-absorbing secondary organic aerosols from toluene

photooxidation, Sci. Total Environ., 794, 148714, 10.1016/j.scitotenv.2021.148714, 2021.

Malecha, K. T., and Nizkorodov, S. A.: Photodegradation of secondary organic aerosol particles as a source of small, oxygenated volatile organic compounds, Environ. Sci. Technol., 50, 9990-9997, 10.1021/acs.est.6b02313, 2016.

Mao, J. Q., Paulot, F., Jacob, D. J., Cohen, R. C., Crounse, J. D., Wennberg, P. O., Keller, C. A., Hudman, R. C., Barkley, M. P., and Horowitz, L. W.: Ozone and organic nitrates over the eastern United States: Sensitivity to isoprene chemistry, J Geophys Res-Atmos, 118, 11256-11268, 10.1002/jgrd.50817, 2013.

Ng, N. L., Kroll, J. H., Chan, A. W. H., Chhabra, P. S., Flagan, R. C., and Seinfeld, J. H.: Secondary organic aerosol formation from m-xylene, toluene, and benzene, Atmos. Chem. Phys., 7, 3909-3922, 10.5194/acp-7-3909-2007, 2007.

O'Brien, J. M., Shepson, P. B., Muthuramu, K., Hao, C., Niki, H., Hastie, D. R., Taylor, R., and Roussel, P. B.: Measurements of alkyl and multifunctional organic nitrates at a rural site in Ontario, J Geophys Res-Atmos, 100, 22795-22804, 10.1029/94jd03247, 1995.

Pathak, R. K., Stanier, C. O., Donahue, N. M., and Pandis, S. N.: Ozonolysis of alpha-pinene at atmospherically relevant concentrations: Temperature dependence of aerosol mass fractions (yields), J Geophys Res-Atmos, 112, Artn D03201, 10.1029/2006jd007436, 2007.

Prinn, R. G., Weiss, R. F., Miller, B. R., Huang, J., Alyea, F. N., Cunnold, D. M., Fraser, P. J., Hartley, D. E., and Simmonds, P. G.: Atmospheric trends and lifetime of $CH_3CCl_3$ and global OH concentrations, Science, 269, 187-192, 10.1126/science.269.5221.187, 1995.

Qi, X., Zhu, S., Zhu, C., Hu, J., Lou, S., Xu, L., Dong, J., and Cheng, P.: Smog chamber study of the effects of NOx and $NH_3$ on the formation of secondary organic aerosols and optical properties from photo-oxidation of toluene, Sci. Total Environ., 727, 138632, 10.1016/j.scitotenv.2020.138632, 2020.

Sarrafzadeh, M., Wildt, J., Pullinen, I., Springer, M., Kleist, E., Tillmann, R., Schmitt, S. H., Wu, C., Mentel, T. F., Zhao, D. F., Hastie, D. R., and Kiendler-Scharr, A.: Impact of NOx and OH on secondary organic aerosol formation from β-pinene photooxidation, Atmos. Chem. Phys., 16, 11237-11248, 10.5194/acp-16-11237-2016, 2016.

Xu, L., Moller, K. H., Crounse, J. D., Kjaergaard, H. G., and Wennberg, P. O.: New Insights into the Radical Chemistry and Product Distribution in the OH-Initiated Oxidation of Benzene, Environ. Sci. Technol., 54, 13467-13477, 10.1021/acs.est.0c04780, 2020.

Yang, Z., Tsona, N. T., Li, J., Wang, S., Xu, L., You, B., and Du, L.: Effects of NOx and $SO_2$ on the secondary organic aerosol formation from the photooxidation of 1,3,5-trimethylbenzene: A new source of organosulfates, Environ. Pollut., 264, 114742, 10.1016/j.envpol.2020.114742, 2020.

Zhang, X., Cappa, C. D., Jathar, S. H., McVay, R. C., Ensberg, J. J., Kleeman, M. J., and Seinfeld, J. H.: Influence of vapor wall loss in laboratory chambers on yields of secondary organic aerosol, Proc. Natl. Acad. Sci. U. S. A., 111, 5802-5807, 10.1073/pnas.1404727111, 2014.

Zhang, X., Schwantes, R. H., McVay, R. C., Lignell, H., Coggon, M. M., Flagan, R. C., and Seinfeld, J. H.: Vapor wall deposition in Teflon chambers, Atmos. Chem. Phys., 15, 4197-4214, 10.5194/acp-15-4197-2015, 2015.

Zhao, D. F., Schmitt, S. H., Wang, M. J., Acir, I. H., Tillmann, R., Tan, Z. F., Novelli, A., Fuchs, H., Pullinen, I., Wegener, R., Rohrer, F., Wildt, J., Kiendler-Scharr, A., Wahner, A., and Mentel, T. F.: Effects of NOx and $SO_2$ on the secondary organic aerosol formation from photooxidation of α-pinene and limonene, Atmos. Chem. Phys., 18, 1611-1628, 10.5194/acp-18-1611-2018, 2018.

Zou, Y., Deng, X. J., Zhu, D., Gong, D. C., Wang, H., Li, F., Tan, H. B., Deng, T., Mai, B. R., Liu, X. T., and Wang, B. G.: Characteristics of 1 year of observational data of VOCs, NOx and $O_3$

at a suburban site in Guangzhou, China, Atmos. Chem. Phys., 15, 6625-6636, 10.5194/acp-15-6625-2015, 2015.

**Response to Reviewers**

Ms. Ref. No.: acp-2021-560

Synergetic effect of NH$_3$ and NOx on the production and optical absorption of secondary organic aerosol formation from toluene photooxidation

Dear Editor:

We greatly appreciate the time and effort that the editor and reviewer spent in reviewing our manuscript. After reading the comments from the reviewers, we have carefully revised our manuscript. All the changes we made are marked in red. Our responses to the comments are itemized below. The referee's comments are in black, authors' responses are in blue.

Anything for our paper, please feel free to contact me via ghwang@geo.ecnu.edu.cn.

All the best

Gehui Wang

Oct.14, 2021

**Reviewer #3**

In this study, the authors reported the synergetic effect of NH$_3$ and NOx on the production and optical absorption of toluene-derived SOA characterized using the AMS. The presence of NH$_3$ and NO$_2$ has different influence on the yield concentration.

Comments:

1. Does the lowest concentration in Exp. 4 (adding $NO_2$) suggest the current reduction of NOx would lead to higher SOA? How to link the discovery of the chamber study to the real environment?

**Author reply:**

In the polluted atmosphere, the formation of $O_3$ and OH was mainly from the photolysis of NOx. SOA formation was not only affected by the NOx but also by the oxidant concentration. Previous studied have pointed a clear increase at first and then a decrease in the SOA yield was found with increasing NOx concentration ,which was consistent with the change trend of OH concentration (Sarrafzadeh et al., 2016; Yang et al., 2020). SOA yield was decreased with increasing NOx concentration after [OH] adjustment.

Toluene was dominantly oxidized by OH in the atmosphere. In this study, OH concentration was almost consistent between each experiment. How NOx effect OH concentration, especially in the real atmosphere, was not consider here. Because the real atmosphere is a mixed system, and a synergetic promotion of $NH_3$ and NOx on SOA formation was observed in this study. Although SOA formation is inhibited when only NOx is present, our result highlights the contribution of compound pollution conditions to SOA formation, and has certain guiding significance for both NOx and $NH_3$ emission reduction.

2. The authors provided several possible causes for the SOA mass variation with the addition of $NH_3$ or $NO_2$ based on literature. It would provide more new information if the authors could quantify the potential mechanism contribution since a detailed AMS analysis was conducted.

**Author reply:**

We are very grateful to the reviewer for this comment.

As far as we know, the study of the synergistic effect of NOx and $NH_3$ on toluene SOA formation is rarely seen. We not only point out the synergistic effect of NOx and $NH_3$ on SOA formation, but also discuss the mechanism of this synergistic effect based on AMS results. Quantitative analysis of synergistic effect of NOx and $NH_3$ on SOA

formation has been investigating in our group, and the quantitative results would be shown later.

Does the determined organonitrogen-related species in AMS reflect the yield concentration change?

**Author reply:**

The N/C ratio is very low, we don't think organonitrogen-related species in AMS reflect the change of SOA concentration.

In addition to the proposed mechanism to the influence of $NO_2$ in the manuscript, could $NO_2$ reacting with OH to form $HNO_3$? Such reaction will provide additional OH consumption and reduces the oxidation level of organic species. Because of the high volatility of $HNO_3$, the overall mass might decrease. The addition of $NH_3$ can further cause the formation of $NH_4NO_3$, which can condense on the particles to increase secondary aerosol formation.

**Author reply:**

We are very grateful to the reviewer for this comment. We measured the toluene concentration with PTR-MS on-line. The evolution of toluene concentration at different experiment conditions was shown below. The evolution of toluene concentration was almost uniformly at different NOx and $NH_3$ conditions.

[Figure]

Fig.S1    The evolution of toluene concentration for each experiment."

The OH concentration in the chamber was calculated based on the first order decay of toluene concentration. There was no obvious difference of OH concentrations in the different NOx and NH$_3$ conditions. The evolution of OH concentration at experiment conditions was shown in Fig.S2. The highest OH concentration of $1.0 \times 10^8$ molecule cm$^{-3}$ was observed at the beginning of the reaction. The average OH concentration over the entire reaction period is $5.9 \times 10^7$ molecule cm$^{-3}$. The different experiment conditions in this study did not affect the OH concentration obviously. Therefore, in this study, the different SOA mass concentration formed under different experimental conditions were not caused by the change of OH concentration.

[Figure]

Fig.S2 The evolution of OH concentrations at different experiment conditions"

The concentration variations of $NO_3^-$ and $NH_4^+$ during the experiments were measured by the AMS. The content of $NO_3^-$ and $NH_4^+$ is very low in particulate matter. The content of $NO_3^-$ and $NH_4^+$ for each experiment was list in the table below.

**Table S1.** The content of $NO_3^-$ and $NH_4^+$ in the particle-phase for each experiment

|  | $[NOx]_0$ (ppb) | $[NH_3]_0$ (ppb) | $[NO_3^-]/[Org]$ (%) | $[NH_4^+]/[Org]$ (%) |
|---|---|---|---|---|
| Exp.2 | - | ~200 | - | 1.9 |
| Exp.3 | 62 | ~200 | 4.0 | 2.6 |
| Exp.4 | 63 | - | <0.2 | - |

As seen in Table S1, when only $NO_2$ was introduced into the chamber, the content of $NO_3^-$ in particulate matter was less than 0.2%. Based on the consumption of $NO_2$, most of the $NO_3^-$ was present in the gas phase in the form of nitric acid. When only $NH_3$ was introduced in the chamber, the content of $NH_4^+$ was about 1.9%. When both $NO_2$ and $NH_3$ were simultaneously introduced into the chamber, the $NO_3^-$ reacted with $NH_4^+$ to form ammonium nitrate particles, which accounted for 6.6% of the total mass of particulate matter in the chamber. AMS cannot distinguish whether $NO_3^-$ and $NH_4^+$ come from organic or inorganic phase, but 6.6% is obviously the upper limit of inorganic components in particulate matter if we assume that both are entirely derived from inorganic phase. The particle mass increased about 59% in the $NH_3$ + NOx experiment compared that with no $NH_3$ or NOx, which cannot be explained by the formation of inorganic $NH_4NO_3$ but can only be ascribed to the synergetic effect of

NH$_3$ and NOx on the toluene SOA formation. The following sentences were added in line 253 of the manuscript to explain the possible influence of the smaller amount of the inorganic aerosol on the enhancement in particle mass in the NH$_3$ + NOx experiment.

"Although inorganic aerosol was formed from the interaction of NH$_3$ and NOx in the chamber, the upper limit of the inorganic matter only account for 6.6% of the total mass of particulate matter (Table S1) in the NH$_3$ + NOx experiment. Therefore, it was not the main cause of the increase in particulate matter."

The y-axes labels of Figure S4 (a) (c) (e) are not in the correct position.

**Author reply:**

Yes, suggestion taken.

**Response to Reviewers**

Ms. Ref. No.: acp-2021-560

Synergetic effect of NH$_3$ and NOx on the production and optical absorption of secondary

organic aerosol formation from toluene photooxidation

Dear Editor:

We greatly appreciate the time and effort that the editor and reviewer spent in reviewing our manuscript. After reading the comments from the reviewers, we have carefully revised our manuscript. All the changes we made are marked in red. Our responses to the comments are itemized below. The referee's comments are in black, authors' responses are in blue.

Anything for our paper, please feel free to contact me via ghwang@geo.ecnu.edu.cn.

All the best

Gehui  Wang

Oct.14, 2021

The authors reported experimental results of secondary organic aerosol (SOA) formation from oxidation of toluene in the absence or presence of NOx and NH3, or both. They used aerosol mass spectrometry (AMS) and UV/vis spectrometry to characterize the chemical composition and light-absorption property, respectively, of the SOA formed under those conditions. They suggested that the presence of both NOx and NH3 had synergetic effects in both SOA formation and the light-absorption ability. Based on literature and their own indirect results from AMS, they suggested that the efficient formation of secondary imines, which are probably cyclic and aromatic, were responsible for the enhanced SOA formation and light absorption. The work is of a clear objective and of interest to the atmospheric chemistry community, and the data obtained supported the conclusion made. But I do have a few concerns, which

are outlined below. The manuscript requires further editing as I found some contradicting descriptions and also difficulty in fully understand the details. I recommend Major Revision before publication in ACP.

Major:

1. The first concern is the high concentrations of toluene used and SOA resulted. Even though atmospheric relevancy is always the concern for smog chamber studies, the characterization methods of this study should allow lower concentrations. The AMS should be able to measure SOA in 10s of microgram per cubic meter quite satisfactorily; a rough calculation of the dissolved SOA in methanol (500 microgram per cubic meter, 3 cubic meter, 5 mL methanol) gives a concentration of 3000 milligram per liter, which should be more than enough to get good UV/vis spectra even if only a portion of the SOA is light-absorbing (if this portion is extremely small, then it is not important). Therefore, there is room to reduce the concentrations for the results to be more atmospherically relevant, e.g., for the gas-phase and heterogeneous chemistry as emphasized by the authors.

**Author reply:**

We thank the reviewers for the very important comments.

As calculated, the dissolved SOA in methanol is about 300 milligram per liter, not 3000 here. According to the actual measurement of the collected particulate matter mass, the dissolved SOA is about 100~150 milligram per liter.

As we reply to the comments in RC#2, atmosphere environment is very complicate and the experimental conditions in the chamber cannot be the same as the real environment. Although the VOCs concentration our study is high, the proportion of the key components is comparable to the ambient level. The mean concentration of aromatics VOCs and OH was about 11 ppb (Zou et al., 2015) and $1 \times 10^6$ molecule cm$^{-3}$ (Prinn et al., 1995), respectively, which was consistent with the VOCs/OH ratio (~700 ppb/$5.9 \times 10^7$ molecule cm$^{-3}$) in our study. Therefore, our results still have some guidance for actual atmospheric reactions.

To provide a better illustration of the experiment condition, the following sentences have been added in the revised manuscript in line 170:

"In our work, the OH and toluene concentrations were higher than those of urban conditions. The purpose of the high OH and toluene concentrations is to obtain enough particle production samples for off-line collections and accurate measurements. The toluene concentrations remained stable under the different experimental conditions, the variation of toluene-derived SOA mass concentration and yield was only affected by the different $NO_2$ and/or $NH_3$ concentrations in this study. Toluene was studied here as the representative of total aromatic VOCs in the urban atmosphere. The concentration ratio of toluene to OH in this study is similar to that under the real atmospheric conditions (Zou et al., 2015; Prinn et al., 1995)."

The follow sentences were also added in line 523 of the revised manuscript:

"It has to be noted that the concentration of reactants used for the experiments is much higher than that observed in polluted areas, the effect of $NH_3$ and NOx on the photooxidation of toluene with lower concentrations would be checked in the further study."

2.  There is a lack in the conditions for the smog chamber experiments. For instance, what was the estimated OH concentration, or OH exposure/equivalent photochemical age, of the experiments? What was the approximate NO/NO2 ratio given the OH conditions and NO2 introduced? Is ozone expected to form in significant amount after introducing toluene, or was it measured? If so, would that also convert NO2 to NO3 and become another different type of oxidant, and might also lead to the formation of light-absorbing products as well? They authors mentioned in L146 that some details have been described in previous studies. But it would good to include citations here, and better yet to provide some critical details for the readers to apprehend the conditions for the smog chamber experiments.

**Author reply:**

We are very grateful to the reviewer for this comment. We measured the toluene concentration with PTR-MS on-line. And the OH concentration was calculated based on the on the first order decay of toluene. In the revised manuscript, the following sentences were added in line 163 "The OH concentration in the chamber was calculated

based on the first order decay of toluene concentration. There was no obvious difference of OH concentrations in the different NOx and $NH_3$ levels (Fig. S2)."

The highest OH concentration of $1.02 \times 10^8$ molecule $cm^{-3}$ was observed at the beginning of the reaction. The average OH concentration over the entire reaction period was $5.87 \times 10^7$ molecule $cm^{-3}$. The calculation of OH concentration was added in the Supporting Information as below:

**"S1 OH Concentration Calculation Process**

The OH concentration was calculated based on the decay ratio of toluene concentrations and the known rate constant with respect to OH. The change of toluene concentration over time can be expressed as:

$$-\frac{d[toluene]}{dt} = K_{OH} \times [OH] \times [toluene] \tag{RS1}$$

Where, $K_{OH}$ is the reaction rate constant of OH radicals with toluene ($K_{OH}=5.7\times 10^{-12}$ $cm^3$ molecule$^{-1}$ $s^{-1}$). Assuming that the concentration of hydroxide did not change during the experiment, then we can get:

$$\ln(\frac{[toluene]_0}{[toluene]_t})/t = K_{OH} \times [OH] \tag{RS2}$$

Thus, plotting the variation curve of $\ln([toluene]_0/[toluene]_t)$ vs. time t showed as Fig.S1. The $\ln([toluene]_0/[toluene]_t)$ in Fig.1(b) was not a straight line. This is because the OH is consumed as the reaction goes on. The evolution of OH concentration at experiment conditions was shown in Fig.S2. The different experiment conditions in this study did not affect the OH concentration obviously. The highest OH concentration of $1.0 \times 10^8$ molecule $cm^{-3}$ was observed at the beginning of the reaction. The average OH concentration over the entire reaction period is $5.9 \times 10^7$ molecule $cm^{-3}$.

[Figure]

Fig.S2 The evolution of OH concentrations at different experiment conditions"

In this study, 254 nm UV light was used for the $H_2O_2$ photolysis and OH formation. The photolysis of $NO_2$ requires 365 nm black light as the light source. We cannot exclude small amount $O_3$ formation from the photolysis of $NO_2$. $O_3$ concentration observed in Exp. 3 and 4 was lower than 10 ppb. $O_3$ does not react with toluene. The formation ratio of $NO_3$ through $O_3$ with $NO_2$ is very slow, only $3.5 \times 10^{-17}$ $cm^3$ molecule$^{-1}$ s$^{-1}$. And the reaction rate of $NO_3$ with toluene is also very low, which is $2\sim6 \times 10^{-17}$ $cm^3$ molecule$^{-1}$ s$^{-1}$. At the same time, $NO_3$ has a short lifetime under the light conditions because $NO_3$ is extremely easy to photolysis. Obviously, $NO_3$ oxidation of toluene is negligible here.

The previous study was included here "Liu S. J., Wang Y., Wang G., et al. Enhancing effect of $NO_2$ on the formation of light-absorbing secondary organic aerosols from toluene photooxidation [J]. Sci. Total Environ., 2021, 794: 148714."

3.    The description of the AMS measurements is a bit ambiguous. First, I think there is some inaccuracy in L194: an HR-ToF-AMS normally does not contain a quadrupole, at least not as a mass analyzer. Second, it was stated that the V mode was used to achieve high signal-to-noise ratio, but there are lots of descriptions about elemental analysis (O/C, H/C, OM/OC, OSC etc.) later. The question is, were W-mode data acquired together with V-mode data for the high-resolution (HR) analysis or the HR fitting was based on V-mode data? Third, what was the purpose of the Nafion dryer if the RH in all experiments did not exceed 27% (Table 1)? Did it significantly reduce particle

bouncing, as compared to the complication of potential particle loss passing the Nafion dryer?

**Author reply:**

Here is a mistake, "quadrupole" has been deleted.

Elemental analysis (O/C, H/C, OM/OC, OSC etc.) later was based on the HR fitting through V-mode data.

We are very grateful to the reviewer for this comment. Nafion dryer is one part of the AMS. The RH was low in this study, but SOA formation with different RH was also be studied in our group. The potential particle loss passing the Nafion dryer is very low (Eatough et al., 1999). Nafion dryer did not cause the potential loss of SOA.

4.     About the SOA mass. First, were the mass concentrations in Figure 1 from SMPS data or AMS data? If the latter, were there any ammonium signals from the AMS spectra in the presence of NH3, and were they "counted" as SOA if any? Maybe whether to call the ammonium ion formed with carboxylic acids (simply neutralization) is a trivial question, but the fragments of m/z 15, 16, 17, 18 etc., which are assigned as inorganic ammonium by default, might come from imines in this study.

**Author reply:**

Mass concentrations in Fig. 1 were obtained from SMPS data.

$NH_4^+$ and $NO_3^-$ are inevitably counted in the SOA mass concentration. The concentration variations of $NO_3^-$ and $NH_4^+$ during the experiments were measured by the AMS. The content of $NO_3^-$ and $NH_4^+$ is very low in particulate matter. The content of $NO_3^-$ and $NH_4^+$ for each experiment was list in the table below.

**Table S1.** The content of $NO_3^-$ and $NH_4^+$ in the particle-phase for each experiment

|        | $[NOx]_0$ (ppb) | $[NH_3]_0$ (ppb) | $[NO_3^-]/[Org]$ (%) | $[NH_4^+]/[Org]$ (%) |
|--------|-----------------|------------------|----------------------|----------------------|
| Exp.2  | -               | ~200             | -                    | 1.9                  |
| Exp.3  | 62              | ~200             | 4.0                  | 2.6                  |
| Exp.4  | 63              | -                | <0.2                 | -                    |

As seen in Table S1, when only $NO_2$ was introduced into the chamber, the content of $NO_3^-$ in particulate matter was less than 0.2%. Based on the consumption of $NO_2$,

most of the $NO_3^-$ was present in the gas phase in the form of nitric acid. When only $NH_3$ was introduced in the chamber, the content of $NH_4^+$ was about 1.9%. When both $NO_2$ and $NH_3$ were simultaneously introduced into the chamber, the $NO_3^-$ reacted with $NH_4^+$ to form ammonium nitrate particles, which accounted for 6.6% of the total mass of particulate matter in the chamber. AMS cannot distinguish whether $NO_3^-$ and $NH_4^+$ come from organic or inorganic phase, but 6.6% is obviously the upper limit of inorganic components in particulate matter if we assume that both are entirely derived from inorganic phase. The particle mass increased about 59% in the $NH_3$ + NOx experiment compared that with no $NH_3$ or NOx, which cannot be explained by the formation of inorganic $NH_4NO_3$ but can only be ascribed to the synergetic effect of $NH_3$ and NOx on the toluene SOA formation. The following sentences were added in line 253 of the manuscript to explain the possible influence of the smaller amount of the inorganic aerosol on the enhancement in particle mass in the $NH_3$ + NOx experiment.

"Although inorganic aerosol was formed from the interaction of $NH_3$ and NOx in the chamber, the upper limit of the inorganic matter only account for 6.6% of the total mass of particulate matter (Table S1) in the $NH_3$ + NOx experiment. Therefore, it was not the main cause of the increase in particulate matter."

Minor:

1. L193: "address" to "achieve"? "signal-to-noise" to "signal-to-noise ratio"?

**Author reply:**

Suggestion taken, we corrected them.

2. L277: it might not be good to label the whole SOA as "SV-OOA" just according to the OSc value. And the authors stated in L407 that actually one factor from PMF analysis was very close to LV-OOA based on the O/C ratio.

**Author reply:**

"Notably, all the toluene SOA was characterized as semi-volatile oxygenated organic aerosols (SV-OOA)" was fixed as: "Notably, toluene SOA $OS_C$ values was in the range between -0.5 and 0, which is consistent with that of semi-volatile oxygenated organic aerosols (SV-OOA)."

3. L313: proper subscript for "OSC".

**Author reply:**

Yes, we corrected it.

4. L337-339: this looks like two sentences.

 **Author reply:**

Yes, we rephrased them.

5. L427: NOC and NOA are used to refer to more or less the same thing. Please keep it consistent.

**Author reply:**

We revised all the "NOCs" as "NOA".

6. L509: "high" to "highly".

**Author reply:**

Corrected

7. Fig. 4: the "relative intensity". It seems it is not relatively to the highest peak, nor the total signal. Then what are they relative to as relative intensities?

**Author reply:**

The relative intensity of each m/z mean that it is relative to the total signal. First, we obtained the toluene SOA spectra in both formation stage and stable stage. Then

subtract the formation stage SOA spectra from the stable stage mass spectra, and finally we got the differential spectra of Fig. 4.

8. Fig. 6: the y-axis title uses different notations as Lo-NOA and Hi-NOA in the legend, and the ratio of the two (as indicated by the y-axis title) should not give a total value of 100%. Should it be just "Percentage in total (%)"?

**Author reply:**

Suggestion taken. And Fig. 6 was revised as below.

[Figure]

**Reference**

Eatough, D. J., Obeidi, F., Pang, Y., Ding, Y., Eatough, N. L., and Wilson, W. E.: Integrated and real-time diffusion denuder sample for PM$_{2.5}$, Atmos. Environ., 33, 2835-2844, 10.1016/s1352-2310(98)00326-4, 1999.

Prinn, R. G., Weiss, R. F., Miller, B. R., Huang, J., Alyea, F. N., Cunnold, D. M., Fraser, P. J., Hartley, D. E., and Simmonds, P. G.: Atmospheric trends and lifetime of CH$_3$CCI$_3$ and global OH concentrations, Science, 269, 187-192, 10.1126/science.269.5221.187, 1995.

Zou, Y., Deng, X. J., Zhu, D., Gong, D. C., Wang, H., Li, F., Tan, H. B., Deng, T., Mai, B. R., Liu, X. T., and Wang, B. G.: Characteristics of 1 year of observational data of VOCs, NOx and O$_3$ at a suburban site in Guangzhou, China, Atmos. Chem. Phys., 15, 6625-6636, 10.5194/acp-15-6625-2015, 2015.

---

## Author Response (AR2)

Dear Editor

Many thanks for your important comments, which are very helpful for further improving our manuscript quality. We have carefully read you comments and revised our manuscript based on your comments and suggestions. We also made additional revisions for accurate discussions and statements. Following is our response to your comments.

Anything for our manuscript, please feel free to contact me via ghwang@geo.ecnu.edu.cn

All the best,

Gehui Wang

Oct. 30, 2021

**Comments from the author**:

This manuscript described a very interesting set of results, and the authors have made substantial changes to address reviewers comments. I have some editorial comments to add here, and I consider these requested revisions to be minor.

**Author reply:** We thank you for the comments. See our response below. Line numbers here refer to those in the revised manuscript:

1. Comment: Line 28 and throughout manuscript: "x" in NOx should be in subscripts
**Author reply**: All the "NOx" in the manuscript have been fixed as "$NO_X$".

2. Comment Line 30-33: Information in the abstract should be broadly applicable to the atmosphere. Report yields (which are intensive) rather than mass concentrations (which are specific to the conditions employed by this study only).

**Author reply:** Suggestion taken. SOA mass concentrations are changed as SOA yield in line 30-33, and shown as follows:

"The SOA yield increased from 28.1% in the absence of $NH_3$ to 34.7% in the presence of $NH_3$ but decreased to 19.5% in the presence of $NO_X$. However, the highest SOA yield of 42.7% and the lowest carbon oxidation state ($OS_C$) occurred in the presence of both $NH_3$ and $NO_X$"

3. Comments: Line 35 and throughout manuscript: "precipitate" usually means transitioning from liquid to solid, or rain/snow. "Partition" is more appropriate.

**Author reply:** "precipitate" in line 35, 106, 384 and 391 are fixed as "partition".

4. Comments: Line 101: I suggest replacing "artificial" with "anthropogenic", and "biological" with "biogenic" to denote the source of emission, rather than how the compound was created.

**Author reply:** "artificial" and "biological" in line 101 are fixed as "anthropogenic" and "biogenic", respectively.

5. Comments: Line 124-125: "contributes much to the improvement… compared to $SO_2$ in winter". Does this mean reduction in $NH_3$ contributes more to reduction in $PM_{2.5}$ than reductions in $SO_2$?

**Author reply:** The content of this expression here is inaccurate. $SO_2$ emissions have be significantly reduced, while particle concentrations have decreased less. For clarification, the sentence in line 101 is changed to:

"Indeed, field observation and model simulation have pointed out that the simultaneous control of $NH_3$ emissions in conjunction with $SO_2$ emission is more effective in reducing $PM_{2.5}$ than the process without $NH_3$ emissions control, and $PM_{2.5}$ concentration can be more effectively reduced if $NH_3$ emission is decreased as much as that of $SO_2$ (Erisman and Schaap, 2004)."

6. Comment: Line 153: "one atmospheric pressure" can be replaced with "atmospheric pressure" or "1 atm"

**Author reply:** "one atmospheric pressure" is fixed as "atmospheric pressure".

7. Comment: Line 154: "at all times"

**Author reply:** It is fixed.

8. Comment: Line 155: replace "at last" with "at least"

**Author reply:** It is fixed.

9. Comment: Line 160: I am curious why the zero air generator does not make RH close to zero.

**Author reply:** During the zero air generation process, the allochroic silicagel absorbs water vapor, which can make the RH less than 20% but cannot make it down to zero, because the silicagel absorption cannot remove all the water vapor. In most cases, <20% RH is believed to be dry enough for smog chamber simulation as a dry conditions, because such a low RH condition is less than the deliquescent point of most inorganic salts in the atmosphere. Thus, zero air with a ~20% RH has widely been used for smog chamber experiments.

10. Comment: Line 169: it is interesting that there is no change in OH concentration despite changing NOx. This suggests that OH production is mainly from H2O2 photolysis, rather than HOx recycling by NOx.

**Author reply:** It is due to the nature of the UV-light used in this study, OH production is mainly from $H_2O_2$ photolysis, rather than HOx recycling by NOx.

11. Comment: Line 175: what is a typical ratio of $NO_2$ to NO? was this measured?

**Author reply:** NOx concentration was measured online by the NO-NO_2-NOx analyzer (Model 42C, Thermo Electron Corporation, USA). The typical ratio of $[NO_2]_0$ to $[NO]_0$ is

about 30:1.

The sentence of "NOx concentration was measured online by the NO-NO₂-NOx analyzer (Model 42C, Thermo Electron Corporation, USA)." is added in line 175.

12. Comment: Line 194: (also pointed out by one of the reviewers) SMPS measures electrical mobility diameter, not "aerodynamic equivalent sizes"

**Author reply:** "aerodynamic equivalent sizes" is fixed as "electrical mobility diameter".

13. Comment: Line 210: (also pointed out by one of the reviewers) V-mode with mass resolution of 2000 might not be sufficient to separate many isobars, especially when N is included. Can the authors show how well isobars are resolved with this resolution? (W-mode, or V-mode in more recent AMS can achieve mass resolution of 4000, which is more capable of quantifying O/C ratios)

**Author reply:** We express our appreciation to the editor for this comment. Here are some HR fit with N or without N (see the detailed figures below). As shown in the right column below, if the N fragments are not added in the RH fit, there is a remarkable difference between the signal of AMS (black scatter line) and fitting result (blue line). However, when N fragments are added into the HR fit process, the fitting blue lines are almost entirely overlapping with the AMS signal. In addition, clearly suggesting that in this study the V-mode of AMS with mass resolution of 2000 is capable of separating many isobars including the N-containing organic fragments, and the related results shown by this work are reliable. Similar work has been reported by other AMS work (Liu et al., 2015) and are referenced by this work.

[Figure]

m/z 87

m/z 59

14. Comment: Line 232: (based on response to one of the reviewers) 46.2 mm, not 46.2 nm

**Author reply:** "46.2 nm" is fixed as "46.2 mm" .

15. Comment: Line 262: this "effect" not "effects"

**Author reply:** This is fixed in the manuscript.

16. Comment: Lines 267-274: this is a thoughtful response to the reviewer's comment. I suggest rephrasing Lines 267-268 because I don't think the wall loss "problem" is necessarily remedied, especially since the authors reported the interesting phenomenon of accelerated wall loss with NH3 present. I think this phenomenon is now buried in the wall loss correction, and not necessarily understood. The authors seem to suggest that carbonyls in the gas phase are consumed by and reacting with NH3 on the walls (which is entirely plausible), causing carbonyls to leave the suspended particles (to maintain equilibrium). If that is the case, is there any evidence from the AMS, or from the size distribution measured by SMPS (e.g. shifting to smaller diameters after lights are turned off)

**Author reply:** The sentences of "Interestingly, wall loss is increased 66% and 205% in Exp.2 (in the presence of NH$_3$) and Exp.3 (in the mixed condition of NH$_3$ and NO$_X$), respectively, when compared with the experiments with no NH$_3$ (Exp.1 and 4). The larger particle wall loss in the presence of NH$_3$ could be explained by increasing condensation process of oxidized organic vapors onto the Teflon chamber wall via oligomerization (for dicarbonyls) and ionic dissociation/acid-base reaction (for organic acids)." in line 267-274 is deleted.

17. Comment: Line 296: this is a good response to the reviewer's concern about NH$_4$NO$_3$ being the source of enhancement. It might be best to rephrase this as what % of the enhancement this source can explain. Also, the author can look at the NH4+ and NO3- signal on the AMS and see how much NH4NO3 is formed.

**Author reply:** We thank the editor for this concern. Our AMS data showed that NH$_4$NO$_3$ formed in he chamber accounted for 6.6% of the SOA mass formed in the chamber.

Therefore, the enhanced particle mass in the chamber are almost entirely due to the SOA formation. Please see the related discussions in page 15, line 294-297.

18. Comment: Line 317-318: this is not a fair statement. SOA aging has received a lot of attention.

**Author reply:** We deleted the sentence of "but little attention has been paid to the evolution of SOA chemical composition in previous studies" in line 317-318.

19. Comment: Line 361: Avoid starting a sentence with "Or".

**Author reply:** Suggestion taken, "Or" is deleted, the manuscript here was fixed as "In addition, Or $NH_3/NH_4^+$ may react with… …".

20. Comment: Line 377-378: formation "of" high volatility oxidation products

**Author reply:** This is fixed in the manuscript.

21. Comment: Line 389-390: "This result suggested that although NOx promotes the formation of higher volatility compounds." Incomplete sentence

**Author reply:** We Combine the two sentences of "This result suggested that although $NO_X$ promotes the formation of higher volatility compounds. These higher volatility compounds (e.g. glyoxal) can react with $NH_3$ and partition into the particle-phase, which could contribute to the increase in SOA formation." into one "This result suggested that although $NO_X$ promotes the formation of higher volatility compounds, these higher volatility compounds (e.g. glyoxal) can react with $NH_3$ and partition into the particle-phase, which could contribute to the increase in SOA formation."

22. Comment: Line 424: "extra-consumed" is an awkward compound word.

**Author reply:** "extra-consumed carbonyl" is fixed as "unreacted carbonyl".

23. Comment: Lines 456 and 457: "defend" should be "defined"

**Author reply:** This is fixed in the manuscript.

24. Comment: Line 473-474: "declining" instead of "decline"

**Author reply:** This is fixed in the manuscript.

25: Comment: Conclusions: several reviewers pointed out the atmospheric relevance of these results. I recommend adding a discussion about how these results at high $NH_3$ levels can be extrapolated to lower concentrations found in the atmosphere.

**Author reply:** Suggestion taken. We added the following discussions into the text. Please see page 29, line 584.

"Although the reactant concentrations including $NH_3$ used in this work are much higher than those in the real urban environment, our results are applicable for the polluted urban atmosphere. In the urban atmosphere aromatic VOCs consist of numerous species and their total concentration is much higher than a single species such as toluene. On the other hand, carboxylic acids and carbonyls in the urban polluted atmosphere can be produced

from aromatics and many other species. Therefore, it is reasonable for our smog chamber experiments to use toluene as a single precursor with a concentration much higher than that in the real atmosphere. Although the mechanisms of SOA formed under high precursor concentrations is expected to be the same as that under low concentrations, the kinetics are probably different. Thus, the effect of $NH_3$ and $NO_X$ on the photooxidation of toluene with lower concentrations should be checked in the further study."

26: Comment: Data availability: following up on one of the reviewer's comments, I highly recommend uploading the data to an online repository. While AMS raw data files are large and it might not be feasible to do so, at the very least all the data used to make the plots should be shared. This sharing will facilitate data comparisons will increase the impact of this study.

**Author reply:** The datasets are available from https://doi.org/10.6084/m9.figshare.1 6910953.

**References:**

Erisman, J. W., and Schaap, M.: The need for ammonia abatement with respect to secondary PM reductions in Europe, Environ. Pollut., 129, 159-163, 10.1016/j.envpol.2003.08.042, 2004.

Liu, Y. C., Liggio, J., Staebler, R., and Li, S. M.: Reactive uptake of ammonia to secondary organic aerosols: kinetics of organonitrogen formation, Atmos. Chem. Phys., 15, 13569-13584, 10.5194/acp-15-13569-2015, 2015.